# SPT6 promotes epidermal differentiation and blockade of an intestinal-like phenotype through control of transcriptional elongation

Jingting Li [1], Xiaojun Xu[2], Manisha Tiwari[1], Yifang Chen[1], Mackenzie Fuller[3,4], Varun Bansal[1], Pablo Tamayo[2,5], Soumita Das[4], Pradipta Ghosh[3] & George L. Sen [1]✉

In adult tissue, stem and progenitor cells must tightly regulate the balance between proliferation and differentiation to sustain homeostasis. How this exquisite balance is achieved is an area of active investigation. Here, we show that epidermal genes, including ~30% of induced differentiation genes already contain stalled Pol II at the promoters in epidermal stem and progenitor cells which is then released into productive transcription elongation upon differentiation. Central to this process are SPT6 and PAF1 which are necessary for the elongation of these differentiation genes. Upon SPT6 or PAF1 depletion there is a loss of human skin differentiation and stratification. Unexpectedly, loss of SPT6 also causes the spontaneous transdifferentiation of epidermal cells into an intestinal-like phenotype due to the stalled transcription of the master regulator of epidermal fate P63. Our findings suggest that control of transcription elongation through SPT6 plays a prominent role in adult somatic tissue differentiation and the inhibition of alternative cell fate choices.

[1] Department of Dermatology, Department of Cellular and Molecular Medicine, UCSD Stem Cell Program, University of California, San Diego, La Jolla, CA, USA. [2] Moores Cancer Center, University of California, San Diego, La Jolla, CA, USA. [3] Departments of Medicine and Cellular and Molecular Medicine, HUMANOID Center of Research Excellence, University of California, San Diego, La Jolla, CA, USA. [4] Department of Pathology, HUMANOID Center of Research Excellence, University of California, San Diego, La Jolla, CA, USA. [5] Division of Medical Genetics, School of Medicine, University of California, San Diego, La Jolla, CA, USA. ✉email: gsen@health.ucsd.edu

Stem and progenitor cells in the basal layer of the epidermis must maintain a tight balance between self-renewal and differentiation to achieve homeostasis[1]. Failure to properly differentiate is the basis for more than 100 human genetic skin diseases that can impact up to 20% of the population[2]. Thus, a tremendous amount of effort has been focused on elucidating the pathways and factors involved in the differentiation process. As basal layer stem and progenitor cells differentiate, they exit out of the cell cycle, detach from the basement membrane and progressively migrate upwards to form the spinous and then the granular layers of the epidermis. Terminal differentiation results in the formation of the stratum corneum which is essential for skin barrier function. Central to this process is the transcription factor P63 which is necessary for both epidermal growth and differentiation with mutations in this gene leading to a variety of skin genetic disorders such as Ectrodactyl-Ectodermal Dysplasia Clefting Syndrome (EEC)[3–7]. Work by our lab and others have also shown a prominent role for the transcription factors ZNF750, KLF4, MAF, MAFB, GRHL1, OVOL1, GRHL3, EHF, and CEBP alpha/beta in promoting differentiation[8–16]. The differentiation program can be controlled at the transcription initiation stage through the recruitment of RNA polymerase II (Pol II) to promoters by specific DNA binding transcription factors. However, there may be additional regulatory steps downstream of transcription initiation such as promoter proximal pausing/transcription elongation that could potentially be a major regulator of the differentiation process[17].

Pol II pausing was first characterized in *Drosophila* heat shock genes and subsequently in the human *MYC* gene where there are much higher levels of Pol II near their promoters than gene bodies suggesting control at the elongation level[18–20]. Pol II can then escape pause and proceed into productive elongation upon the appropriate stimulus such as stress (heat shock genes) or serum (*MYC*). More recent genome-wide studies have found that promoter proximal pausing regulates developmental genes in both *Drosophila* embryos and murine and human embryonic stem cells (ESCs)[21–24]. Actively transcribed genes in ESCs were also subject to promoter-proximal pausing suggesting the importance of this mode of gene regulation[21]. It is currently unclear whether there is any role for control of transcription elongation in adult cell fate transitions such as during somatic tissue differentiation.

Pol II pauses after being recruited to gene promoters by transcription factors on a significant portion of metazoan genes[25]. It pauses within the first 20–60 bp downstream of the transcription start site (TSS) after synthesis of a short nascent RNA and remains paused until receiving further signals for elongation[26]. Promoter paused Pol II is enriched for serine 5 (Ser5) phosphorylation in its C terminal domain (CTD) and bound by the DRB sensitivity-inducing factor (DSIF: SPT4 and SPT5) and negative elongation factor (NELF)[27,28]. Productive elongation occurs when positive elongation factor b (P-TEFb: CDK9 and Cyclin T) phosphorylates DSIF, NELF, serine 2 (Ser2) of Pol II's CTD[29] and CTD linker[30]. Phosphorylated NELF dissociates from the complex which allows Ser2 Pol II, DSIF, PAF complex, and SPT6 to enter in the elongation phase of transcription[30,31]. Loss of NELF binding allows PAF association with Pol II, while CTD linker phosphorylation recruits SPT6[30]. Association with SPT6 and PAF allows conformational changes in the DSIF clamps, which may drive the polymerase forward to promote elongation[30]. Elongation can also be enhanced by association with the super elongation complex (ELL1/2/3, EAF1/EAF2, AFF1/AFF4, AF9/ENL, and pTEFb), which is necessary for the expression of rapidly induced genes[17]. Upon promoter pause escape, the elongating Pol II must also overcome the nucleosome barrier through association with the histone chaperones FACT (SPT16

and SSRP1) and SPT6[32–34]. FACT and SPT6 both have histone chaperone activity allowing them to facilitate the disassembly and reassembly of nucleosomes as Pol II passes through resulting in productive elongation.

SPT6 interacts with both histones H3 and H4 to disassemble and reassemble nucleosomes to allow passage of Pol II during elongation[35,36]. Spt6 has also been found to increase the elongation rate of Pol II on naked DNA and in *Drosophila* S2 cells[37,38]. It can also regulate the expression of lncRNAs through the redistribution of H3K36me3 marks[39]. Spt6 is necessary to promote murine ESC self-renewal by preventing the polycomb repressive complex from silencing super enhancers controlling the expression of pluripotency transcription factors[40]. Surprisingly, whether SPT6 has any role in adult stem cell homeostasis and lineage differentiation is unknown.

Here, we show that epidermal genes, including ~30% of induced differentiation genes already contain paused Pol II in undifferentiated, proliferating epidermal cells, which is released into productive elongation upon differentiation. Through a small RNAi screen of factors involved in transcription elongation, we identify SPT6 as being necessary to promote epidermal differentiation. SPT6 promotes the elongation of genes that encode transcription factors and structural proteins that are essential for the differentiation process. In the absence of SPT6, Pol II is stalled at promoter/TSS regions of differentiation genes which inhibits human skin stratification and differentiation. In the same genetic pathway as SPT6, PAF1 depletion blocks epidermal differentiation due to a loss of transcription elongation of differentiation genes. Unexpectedly, SPT6 loss not only prevents skin stratification but also results in the transdifferentiation of epidermal cells into an intestinal-like phenotype. This is due to Pol II pausing and subsequent loss of expression of the master regulator of epidermal fate, P63, upon SPT6 depletion. Importantly, ectopic expression of P63 in SPT6 depleted cells can block the expression of intestinal genes. Our results suggest that control of transcription elongation through SPT6 plays a prominent role in adult somatic cell fate choices.

## Results

**Promoter proximal pausing and subsequent transcription elongation regulates epidermal differentiation.** To investigate whether Pol II is already bound to differentiation genes in epidermal stem and progenitor cells and how it changes upon induction of differentiation, we performed Pol II ChIP-Seq in primary human keratinocytes cultured in proliferation (subconfluent) and differentiation (confluent with 1.2 mM calcium for 3 days) conditions (Fig. 1a and Supplementary Data 1 and 2). MACS (model-based analysis of ChIP-Seq with cutoff $q$ value of 0.05) was used to call significant peaks in both proliferation and differentiation conditions. There was similar Pol II binding across genic regions in both proliferation and differentiation conditions (Supplementary Fig. 1a). Increased accumulation of Pol II at promoter proximal regions relative to the gene body has previously been used to provide evidence that promoter pausing/transcription elongation is an important mode of gene regulation[21–23,41]. To characterize whether there is promoter proximal pausing, we calculated the travel ratio (TR) of the genes bound by Pol II in proliferation and differentiation conditions. TR is defined as the relative ratio between Pol II density in the gene body compared to the promoter proximal regions (Fig. 1b)[21,41]. A TR of 1 suggests the amount of elongating Pol II is the same as promoter paused Pol II, whereas a TR of 0.25 suggests that there is four times more Pol II paused than elongating. Thus, we have divided these genes into highly (TR ≤ 0.25), moderately (0.25 < TR < 1) and non-paused (TR ≥ 1), according to their TR values.

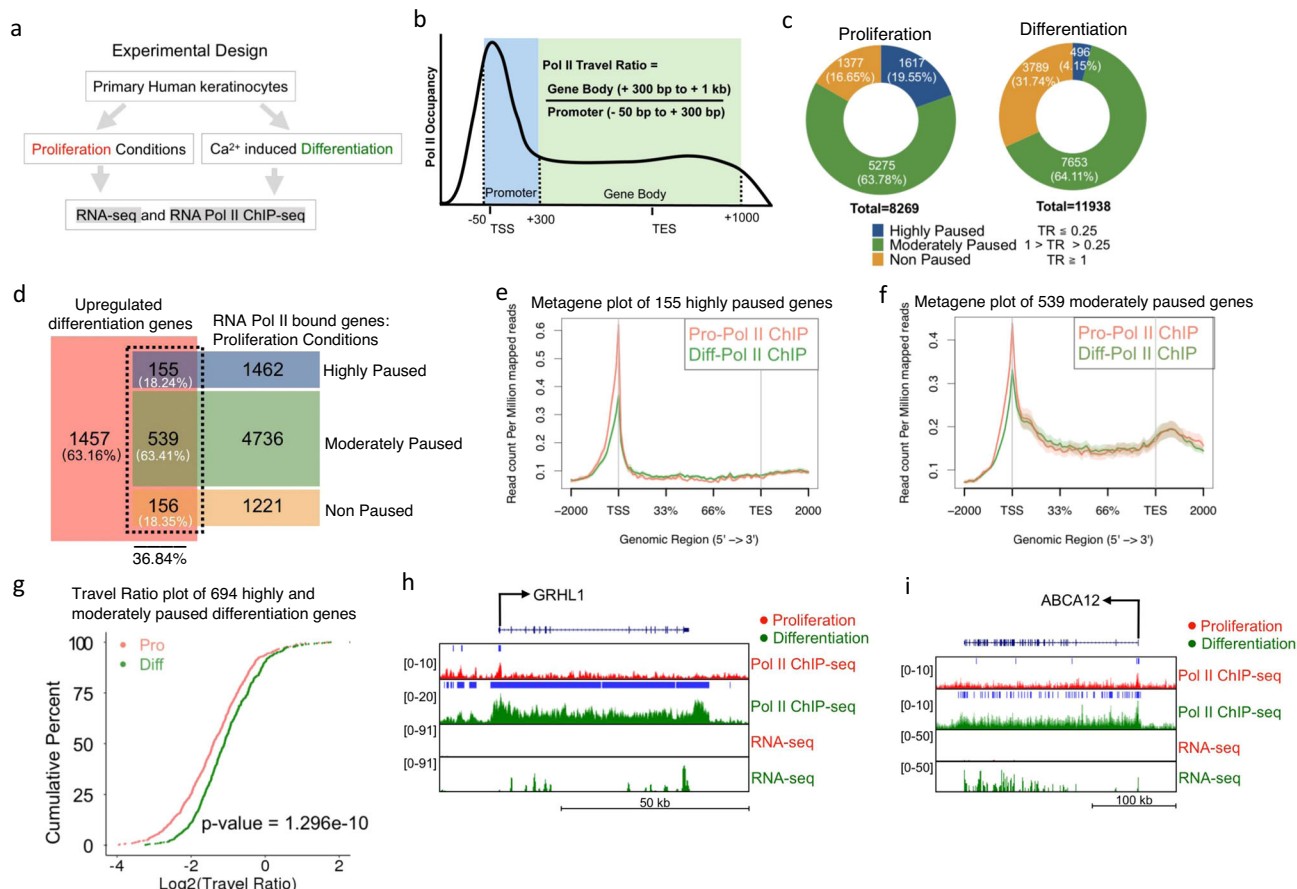

**Fig. 1 Promoter proximal pausing controls differentiation gene expression. a** Diagram of the experimental design. Primary human keratinocytes were either grown in proliferation or differentiation conditions and subjected to RNA Pol II ChIP-Seq. **b** Schematic representation describing the calculation of the travel ratio (TR) for genes bound by Pol II. The promoter proximal region is defined as 50 bp upstream to 300 bp downstream of the transcription start site (TSS). The gene body is defined as 300 bp downstream of the TSS to 1 kb downstream of the transcription end site (TES). **c** Donut chart of percentage of genes that are highly paused (TR ≤ 0.25), moderately paused (1 > TR > 0.25), non-paused (TR ≥ 1) in proliferation and differentiation conditions. Genes that contained significant Pol II binding (Pol II ChIP-Seq) in either proliferation or differentiation conditions were subjected to TR analysis as described in **b**. N = 3 independent experiments for RNA Pol II ChIP-Seq in proliferation or differentiation conditions. **d** Overlap of upregulated differentiation genes with RNA Pol II bound genes in proliferation conditions. The Pol II bound genes have been categorized as highly paused (TR ≤ 0.25), moderately paused (1 > TR > 0.25), or non-paused (TR ≥ 1). **e** Metagene plot of the 155 differentiation genes highly paused in proliferation conditions. Pol II ChIP-Seq in proliferation (Pro) conditions is shown in red and compared to differentiation (Diff) conditions (shown in green). Read count per million mapped reads is shown on the Y-axis. The X-axis represents regions across genes. TSS is transcription start site and TES is transcription end site. The plot is an average of the triplicate samples for each group. **f** Metagene plot of the 539 differentiation genes moderately paused in proliferation conditions. Pol II ChIP-Seq in proliferation (Pro) conditions is shown in red and compared to differentiation (Diff) conditions (shown in green). The plot is an average of the triplicate samples for each group. **g** Travel ratio plot of the 694 differentiation genes highly and moderately paused in proliferation conditions. The average TR of genes in proliferation (Pro) conditions are shown in red and differentiation (Diff) conditions are shown in green. The Y-axis shows the cumulative percentage of genes and X-axis shows the travel ratio plotted in Log2. Gene tracks of two highly paused differentiation genes *GRHL1* (**h**) and *ABCA12* (**i**). Pol II ChIP-Seq tracks and RNA-Seq tracks are shown in proliferation (red) and differentiation (green) conditions. Y-axis shows reads per million and X-axis shows regions along the gene. Blue bars over gene tracks show significant binding.

We found that a total of 1617 genes (19.55%) were highly paused, 5275 genes (63.78%) were moderately paused, and 1377 genes (16.65%) were non-paused in proliferating cells (Fig. 1c). Interestingly, in differentiation conditions the percentage of highly paused genes dropped to 4.15% (496 genes), while non-paused genes increased to 31.74% (3789 genes) suggesting that Pol II pausing/transcription elongation may play a role in epidermal differentiation (Fig. 1c). To orthogonally validate our Pol II results, we used a previously published timecourse of Pol II binding during epidermal differentiation (days 0, 2, 4, and 7)[42]. Similar to our Pol II data, highly paused genes decreased during differentiation from 17.73% in undifferentiated cells down to 11.21% by day 4 of differentiation (Supplementary Fig. 1b). There was also an increase in non-paused genes by day 7 of

differentiation (Supplementary Fig. 1b). To address whether Pol II pausing regulates epidermal differentiation, we analyzed (FDR < 0.05 and ≥2-fold change) a previously published RNA-Seq dataset comparing primary human keratinocytes cultured in proliferation and differentiation conditions using the same timepoints as our work[12]. In total, 2307 genes were upregulated during differentiation which were enriched for gene ontology (GO) terms such as epidermis development (Supplementary Fig. 1c, d and Supplementary Data 3). In total, 2129 genes were downregulated during differentiation with enrichment for proliferation related GO terms (Supplementary Fig. 1c, e and Supplementary Data 3). Of the 2307 upregulated epidermal differentiation genes, ~37% (850/2307) of those genes already had Pol II binding in proliferation conditions (Fig. 1d). The high

prevalence of Pol II bound differentiation genes in progenitor cells is not due to their premature differentiation as early differentiation protein K1 is not detectable in proliferation conditions but stains the vast majority of differentiated cells (Supplementary Fig. 1f, g). Approximately 82% (694/850) of the differentiation genes that already had Pol II binding in progenitor cells were either highly paused or moderately paused, while a minority (18.35%) were not paused (Fig. 1d). The vast majority (>95%) of these genes had little to no expression with RPKM values of <50, suggesting that Pol II pausing prevents productive elongation of these genes (Supplementary Fig. 1h). We decided to focus on these differentiation genes that are paused in progenitor cells to see if they are released during differentiation. A plot of the 155 highly paused and the 539 moderately paused genes showed high levels of promoter proximal Pol II pausing in progenitor cells which is dramatically reduced upon differentiation (Fig. 1e, f). Importantly, the estimated cumulative distribution function (ECDF) TR of the 694 genes significantly increased ($p$ value = 1.296e−10) upon differentiation suggesting that ~30% (694/2307) of the induced epidermal differentiation genes are highly to moderately paused in progenitor cells, which is then released into productive elongation upon differentiation (Fig. 1g).

A closer examination of individual differentiation genes such as GRHL1, ABCA12, and OVOL1 showed high levels of promoter proximal Pol II pausing with no or minimal levels of Pol II in the gene bodies of progenitor cells (Fig. 1h, i and Supplementary Fig. 1k). Upon differentiation, there is a dramatic increase in Pol II onto the gene bodies of these genes suggesting regulation on the level of promoter pausing/transcription elongation (Fig. 1h, i and Supplementary Fig. 1k). RNA-Seq tracks show these genes are only expressed upon differentiation (Fig. 1h, i and Supplementary Fig. 1k). The timecourse data for these genes also show paused Pol II in undifferentiated cells which led to progressively greater levels of elongation as differentiation proceeds (Supplementary Fig. 1i, j). GRHL1 and OVOL1 are both transcription factors that are critical for the differentiation process[14,16]. ABCA12 is necessary for the transport of lipids to form the barrier of the skin with mutations in the gene causing harlequin ichthyosis[43]. While ~37% (850/2307) of the induced epidermal differentiation genes already had Pol II binding in progenitor cells, the remaining ~63% (1457/2307) of the genes did not have binding (Fig. 1d). These differentiation genes include KRTDAP, KLK7, and S100A8 which had no Pol II binding "or mRNA expression" in progenitor cells but gained it upon differentiation (Supplementary Fig. 1l–n). These genes are regulated in the "traditional" manner in that Pol II is likely recruited to these genes only upon the induction of differentiation by transcription factors. To test the possibility that the paused differentiation genes were enriched for transcription factors that could potentially recruit Pol II to the remaining ~63% of induced differentiation genes, we took three approaches. First, we performed GO on the 694 paused and 1457 (without Pol II loaded) differentiation genes. The paused differentiation genes were enriched for "regulators of transcription and gene expression", while the other 1457 differentiation genes did not enrich for those terms (Supplementary Fig. 2a, b). Second, we manually looked for transcription factors previously published to promote epidermal differentiation in both gene sets. Interestingly, there were 14 transcription factors found in the highly and moderately paused genes (694 genes) including GRHL1, CEBPG, MAFB, POU2F3, RORA, POU3F1, KLF4, MAF, FOSL2, DLX3, DLX5, CEBPB, KLF5, and OVOL1 (Supplementary Fig. 2c). There were only four transcription factors found in the induced differentiation genes (1457 genes) that did not have Pol II binding in proliferative cells (Supplementary Fig. 2d). Thus, the paused genes had a 3.5-fold enrichment (14 versus 4) in transcription

factors known to be important for epidermal differentiation even though it had 50% less overall genes (694 versus 1457 genes). Third, analysis of the 1457 genes using Enrichr showed that the GRHL family, OVOL1, and KLF5 are co-expressed with these genes[44] (Supplementary Fig. 2e). Collectively, these results suggest that promoter proximal pausing/transcription elongation is an important regulatory step during epidermal differentiation and that transcription factors regulated by pausing may turn on the rest of the differentiation genes without paused Pol II.

**SPT6 is necessary to promote human epidermal differentiation.** Since Pol II promoter pausing occurs at a significant number of differentiation genes in progenitor cells, this suggests that elongation or pause release factors may play a role in the differentiation process. To test this, we performed a small RNAi screen targeting these factors (SPT5, SPT16, SSRP1, SPT6, ELL, ELL2, AFF1, and AFF4) in differentiated human epidermal cells and achieved >75% knockdown of each gene (Fig. 2a). Knockdown efficiency was also checked on the protein level for a couple of the genes such as SPT5 and ELL that showed good depletion (Fig. 2b). SPT6 was the only one that blocked expression of differentiation genes TGM1 and GRHL1 (Fig. 2c,d). These results were validated using two distinct siRNAs targeting SPT6 (Fig. 2a, c, d). To determine the impacts of SPT6 loss in a tissue setting that allows faithful representation of the stratification and gene expression program of human epidermis, control (CTLi) and SPT6 knockdown (SPT6i) primary human keratinocytes were seeded onto devitalized human dermis to regenerate human skin[45,46]. Depletion of SPT6 blocked skin stratification, complete absence of stratum corneum formation, and loss of expression of late and early differentiation genes/proteins FLG, LOR, K10, HOPX, ABCA12, DSG3, and OVOL1 (Fig. 2e–j). Many of these differentiation genes are also paused in progenitor cells including GRHL1, ABCA12, DSG3, and OVOL1. The proliferative capacity of the basal layer was also diminished as shown by the loss of ki67 positive cells which resulted in a hypoplastic tissue (Fig. 2f–i). RNA-Seq was also performed on CTLi and SPT6i tissue to determine the gene expression program that SPT6 regulates. In total, 4275 genes (FDR < 0.01 and ≥2-fold change) were upregulated upon SPT6 knockdown which were enriched in GO terms such as transcription elongation (Fig. 2k, l and Supplementary Data 4). In total, 1465 genes were reduced in expression upon SPT6 loss which were enriched in genes involved in skin development and keratinocyte differentiation (Fig. 2k, m and Supplementary Data 4). Close to 30% of the genes overlapped between the SPT6 and the differentiation gene expression signature ($p < 7.510e−31$) supporting the critical role of SPT6 in promoting differentiation (Fig. 2n). The correlation was even stronger comparing the overlap between the SPT6 signature with either the paused differentiation (~44% overlap: 303/694) or the non Pol II loaded differentiation genes (~38% overlap:559/1457) (Supplementary Fig. 2f, g). This suggests that this second group of genes (1457 non-paused) may be regulated by the differentiation promoting transcription factors that are pause regulated (Supplementary Fig. 2).

**SPT6 binds to induced epidermal differentiation genes and promotes their transcription elongation.** To ascertain whether SPT6 directly regulated the expression of epidermal differentiation genes, we performed ChIP-Seq on SPT6 in differentiated cells to determine where it bound in the genome. SPT6 bound to 20,782 peaks with 83% of the bound sites mapping to genic regions (5′UTR, promoter, intron, exon, TTS, 3′UTR) (Fig. 3a and Supplementary Data 5). The 20,782 peaks mapped back to 6307 genes that were enriched for keratinocyte differentiation and

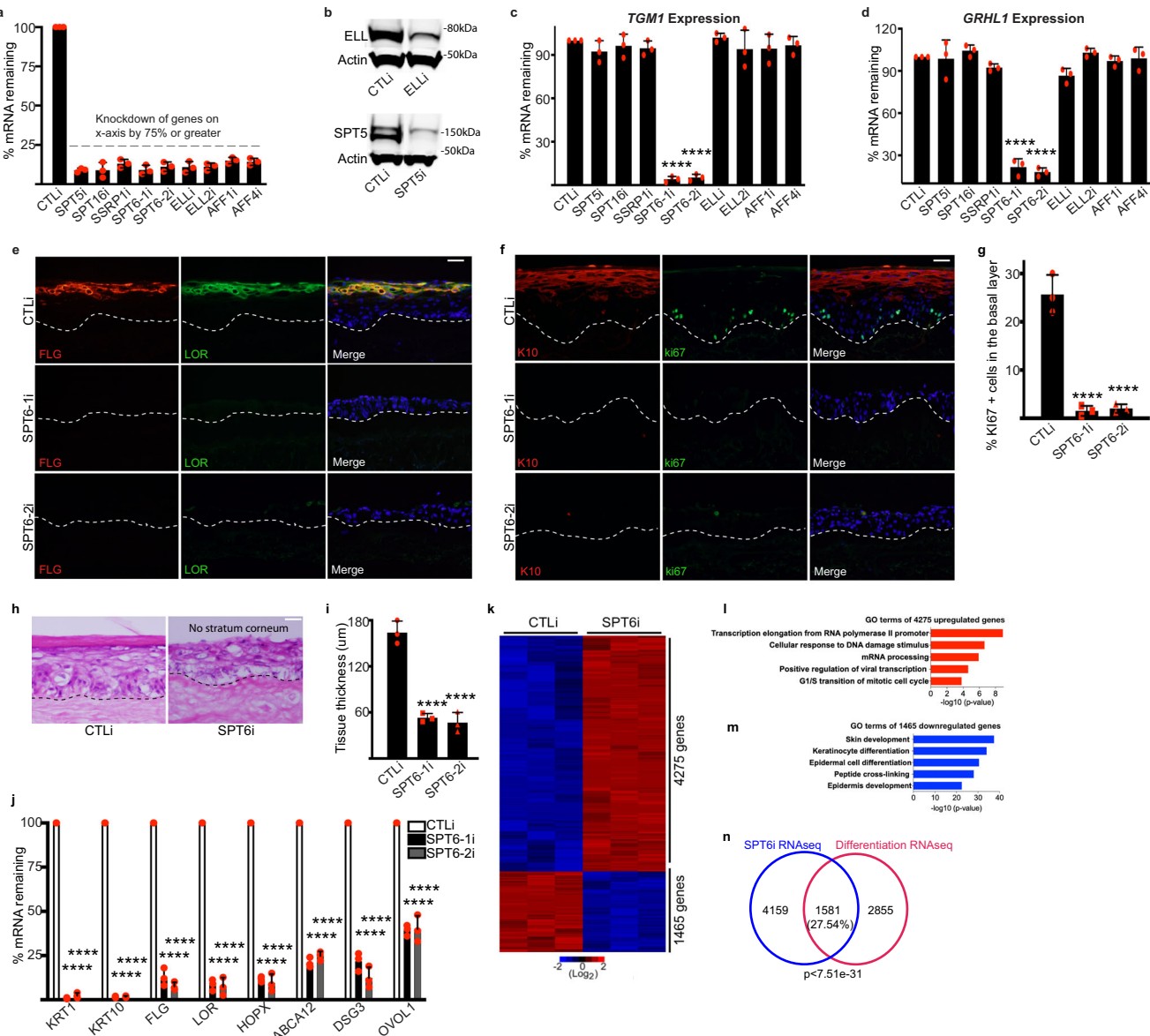

**Fig. 2 SPT6 is necessary for human epidermal differentiation. a** Eight elongation factors were knocked down greater than 75% by siRNAs. Knockdown cells were placed in high confluence and calcium for 3 days to induce differentiation. RT-qPCR was used to quantify the knockdown levels of each elongation factor shown on the *X*-axis. Two separate siRNAs (SPT6-1i and SPT6-2i) targeting different regions of SPT6 were used. qPCR values were normalized to *L32*. *n* = 3 independent experiments. **b** Western blot for knockdown of SPT5 and ELL. *N* = 3 independent experiments which all showed similar results. **c, d** RT-qPCR quantifying the relative mRNA expression levels of epidermal differentiation genes *TGM1* and *GRHL1* in differentiated cells as described in **a**. *n* = 3 independent experiments. **e** Immunofluorescent staining of late differentiation markers FLG (red) and LOR (green) in day 6 regenerated human epidermis treated with control (CTLi) or two distinct siRNAs targeting SPT6 (SPT6-1i and SPT6-2i). Merged image includes Hoechst staining of nuclei. *n* = 3 independent experiments, scale bar = 50 μm. **f** Immunofluorescent staining of early differentiation marker K10 (red) and proliferation marker ki67 (green) in day 6 regenerated human epidermis treated with control (CTLi) or two distinct siRNAs targeting SPT6 (SPT6-1i and SPT6-2i). Merged image includes Hoechst staining of nuclei. *n* = 3 independent experiments. Scale bar = 50 μm. **g** Quantification of the %ki67 positive cells in the basal layer from images taken in **f**. *n* = 3 independent experiments. At least 100 basal layer cells were quantitated per experiment. **h** Hematoxylin and eosin staining of CTLi or SPT6i knockdown regenerated human epidermis (day 6). *N* = 3 independent experiments which all showed similar results. Scale bar = 50 μm. **i** Quantification of the tissue thickness from images taken from **e**, **f**. *n* = 3 independent experiments. **j** RT-qPCR quantifying the relative mRNA expression levels of epidermal differentiation genes in CTL and SPT6 knockdown regenerated human epidermis (day 6). *N* = 3 independent experiments. **k** RNA-Seq analysis of CTL and SPT6 knockdown regenerated human epidermis (day 6). *N* = 3 independent experiments. In total, 4275 genes were upregulated (red) and 1465 genes were downregulated (blue) upon SPT6 depletion. Heatmaps are shown in Log2 scale. **l** Gene ontology (GO) terms for the 4275 genes upregulated in SPT6i tissue using hypergeometric distribution. **m** GO terms for the 1465 genes reduced in expression upon SPT6 loss using hypergeometric distribution. **n** Overlap of the SPT6 knockdown gene expression signature with the differentiation signature. *p* value was calculated using hypergeometric distribution. Mean values are shown with error bars = SD. ****$p < 0.0001$ (two-sided *t*-test for **c**, **d**, **g**, **i**, **j**).

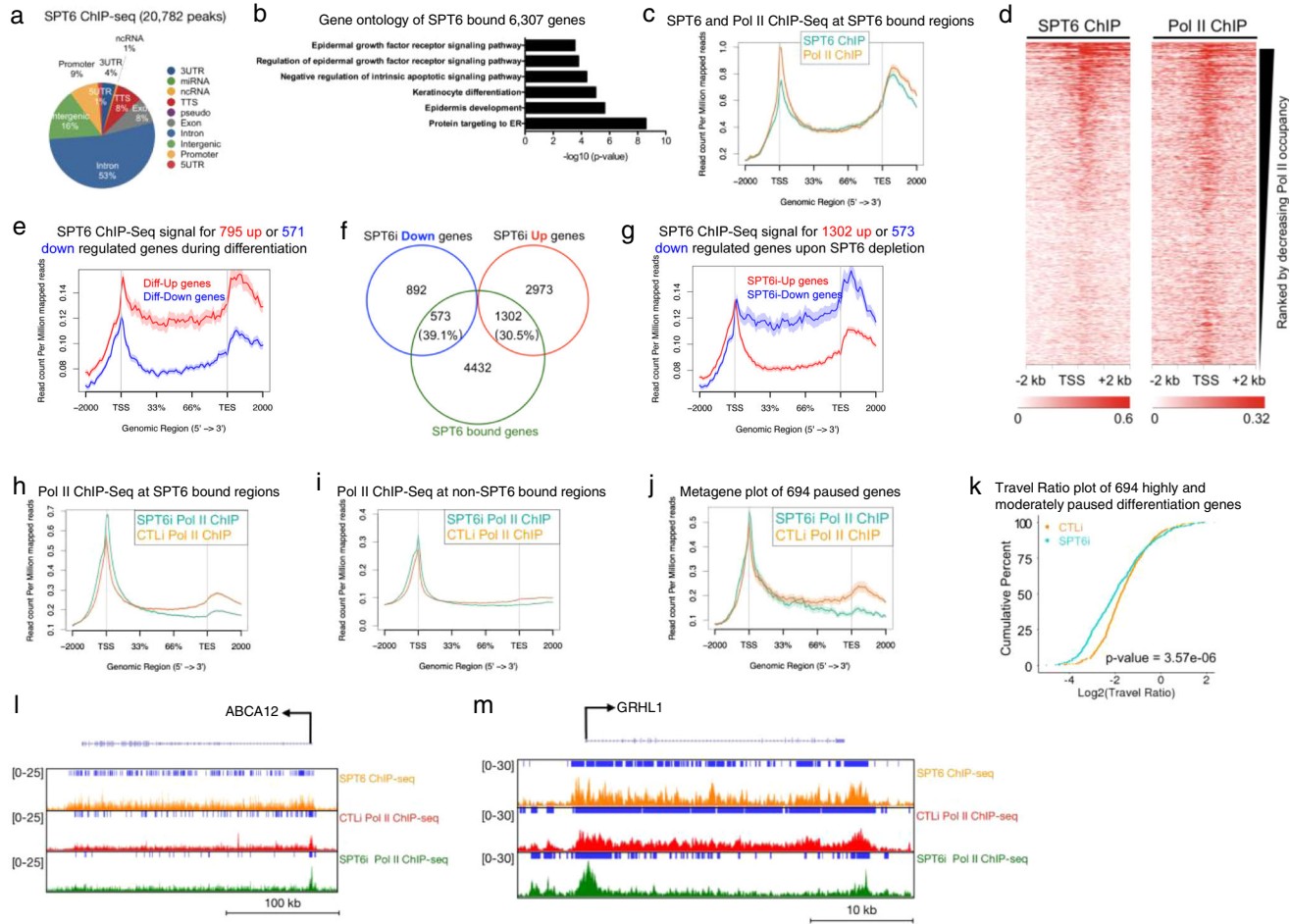

**Fig. 3 SPT6 binds to epidermal differentiation genes and is necessary for their transcriptional elongation. a** Genomic localization of the 20,782 SPT6 bound peaks. SPT6 ChIP-Seq was performed in replicates from day 3 differentiated keratinocytes. **b** Gene ontology terms of the 6307 genes that the 20,782 SPT6 bound peaks mapped back to. **c** Metagene plot of SPT6 (teal) and RNA Pol II (orange) ChIP-Seq reads at SPT6 bound regions. Y-axis is shown as read count per million reads and X-axis is distance along SPT6 bound genes. **d** Heatmap of SPT6 and Pol II ChIP-Seq ranked by decreasing Pol II occupancy. X-axis shows −2 kb to +2 kb from the TSS. **e** SPT6 ChIP-Seq signal for the 795 genes upregulated upon differentiation (Diff-Up) is shown in red. SPT6 ChIP-Seq signal for the 571 genes downregulated upon differentiation (Diff-Down) is shown in blue. Read count per million mapped reads is shown on the Y-axis. The X-axis represents regions across genes. TSS is transcription start site and TES is transcription end site. Average values are shown between experiments. **f** Venn diagram of SPT6 bound genes (SPT6 ChIP-Seq) with genes increased or decreased upon SPT6 depletion. **g** SPT6 ChIP-Seq signal for the 1302 genes upregulated upon SPT6 knockdown (SPT6i-Up genes) is shown in red. SPT6 ChIP-Seq signal for the 573 genes downregulated upon SPT6 loss (SPT6i-Down genes) is shown in blue. Read count per million mapped reads is shown on the Y-axis. The X-axis represents regions across genes. TSS is transcription start site and TES is transcription end site. Average values are shown between experiments. **h** Plot of CTLi (orange) and SPT6i (teal) Pol II ChIP-Seq at SPT6 bound regions. Y-axis is shown as read count per million reads and X-axis is distance along SPT6 bound genes. CTLi and SPT6i Pol II ChIP-Seq in differentiation conditions were performed in replicates. **i** Plot of CTLi (orange) and SPT6i (teal) Pol II ChIP-Seq at non-SPT6 bound regions. **j** Plot of CTLi (orange) and SPT6i (teal) Pol II ChIP-Seq at the 694 differentiation genes paused in proliferation conditions. **k** Travel ratio plot of the 694 differentiation genes paused in proliferation conditions. The average travel ratio of those genes in CTLi (orange) and SPT6i (teal) cells in differentiation conditions is shown. The Y-axis shows the cumulative percentage of genes and X-axis shows the travel ratio plotted in Log2. p value calculated using two-sided Fisher's Exact test. Gene tracks of *ABCA12* (**l**) and *GRHL1* (**m**). SPT6 ChIP-Seq in differentiation conditions is shown in orange. CTLi (red) and SPT6i (green) Pol II ChIP-Seq in differentiation conditions are also shown. Y-axis shows reads per million and blue bar over gene tracks represent significant peaks.

epidermis development GO terms (Fig. 3b and Supplementary Data 5). SPT6 binding to differentiation genes suggests that it may regulate elongation. Supporting this, RNA Pol II's binding profile mimicked SPT6's at SPT6 bound regions (Fig. 3c). Similarly, regions of higher Pol II density also correlated with increased SPT6 binding with a Pearson coefficient of 0.88 (Fig. 3d and Supplementary Fig. 3a). SPT6 bound to ~35% of the genes upregulated during differentiation, while also binding to ~27% of the downregulated genes (Supplementary Fig. 3b). However, SPT6 binding to upregulated differentiation genes was much stronger than downregulated genes (Fig. 3e). Close to 40% (573/

1465) of genes downregulated upon SPT6 loss were bound by SPT6 suggesting direct regulation (Fig. 3f). These 573 genes were also enriched for GO terms such as skin development and keratinocyte differentiation (Supplementary Fig. 3c). SPT6 also bound to genes increased upon SPT6 depletion but binding was significantly less than downregulated genes (Fig. 3f, g). To validate the SPT6 binding data, SPT6 ChIP-QPCR was performed in CTLi and SPT6i cells placed in differentiation conditions. Depletion of SPT6 led to loss of SPT6 binding to differentiation genes such as *ABCA12, TGM1, GRHL1, DSG3,* and *HOPX* (Supplementary Fig. 3d). These results suggest that high levels of SPT6 binding to

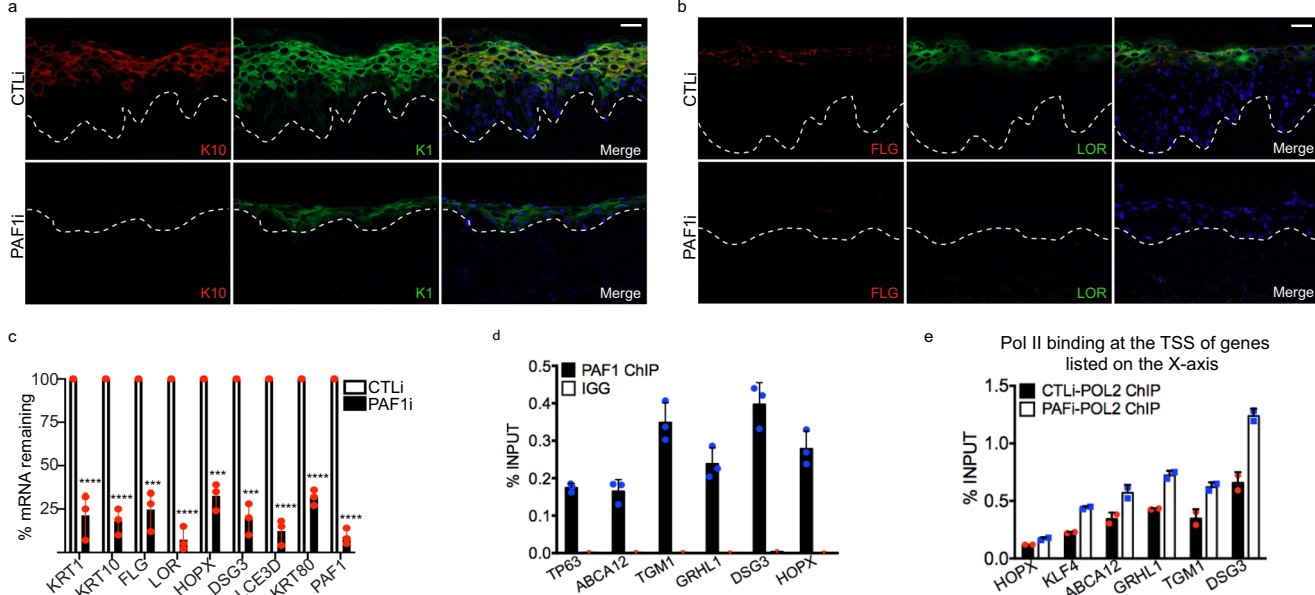

**Fig. 4 PAF1 is necessary for differentiation by promoting transcription elongation. a** Immunofluorescent staining of early differentiation markers Keratin 10 (K10: red) and Keratin 1 (K1: green) in day 6 regenerated human epidermis treated with control (CTLi) or PAF1 siRNAs (PAF1i). Merged image includes Hoechst staining of nuclei. $n = 3$ independent experiments. Scale bar = 50 μm. **b** Immunofluorescent staining of late differentiation markers FLG (red) and LOR (green) in day 6 regenerated human epidermis treated with control (CTLi) or PAF1 siRNAs (PAF1i). Merged image includes Hoechst staining of nuclei. $n = 3$ independent experiments, scale bar = 50 μm. **c** RT-qPCR quantifying the relative mRNA expression levels of epidermal differentiation genes in CTL and PAF1 knockdown regenerated human epidermis (day 6). $N = 3$ independent experiments, two-sided $t$-test. *** = 0.003 for FLG, *** = 0.0001 for HOPX, *** = 0.0001 for DSG3. ****$p < 0.0001$. **d** ChIP-QPCR of differentiation genes in differentiated primary human keratinocytes with PAF1 pulldown. IGG pulldowns were used as a negative control. All primers were targeted toward the transcription start site (TSS) of each differentiation gene. Results are shown as a percent of input. $N = 3$ independent experiments. **e** ChIP-QPCR of RNA Pol II pulldown in both CTLi and PAF1i differentiated cells. Each pulldown was normalized to its respective input and enrichment shown as a percent of input. All primers were targeted toward the TSS of each differentiation gene. $N = 2$ independent experiments. Mean values are shown with error bars = SD.

upregulated differentiation genes may be critical for their expression. To determine how SPT6 impacts Pol II binding, RNA Pol II ChIP-Seq was performed in CTLi and SPT6i cells cultured in differentiation conditions (Supplementary Data 6a, b). At SPT6 bound genes, loss of SPT6 resulted in loss of Pol II from the gene body and accumulation at promoter proximal regions (Fig. 3h). At genes where SPT6 does not bind, there was minimal change to the distribution of Pol II upon SPT6 knockdown demonstrating that SPT6 specifically impacts Pol II distribution at SPT6 bound genes (Fig. 3i). SPT6 knockdown also caused depletion of RNA Pol II from the gene body of the 694 paused differentiation genes and significantly decreased the TR of those genes (Fig. 3j, k). Loss of SPT6 at SPT6 bound differentiation genes such as *ABCA12*, *GRHL1*, and *DSG3* resulted in loss of Pol II from their gene bodies and accumulation at promoter proximal regions (Fig. 3l, m and Supplementary Fig. 3e). These data suggest that SPT6 is necessary for the transcription elongation of its bound epidermal differentiation genes.

**PAF1 promotes epidermal differentiation through control of transcription elongation.** Since SPT6 and the PAF complex have been identified as being part of the core activated elongation complex in vitro, we wanted to determine if PAF performed the same function as SPT6 in a tissue setting. To do this, we knocked down PAF1 and regenerated human skin. Similar to SPT6 loss, PAF1 depletion blocked early and late differentiation genes/proteins K1, K10, FLG, LOR, HOPX, KRT80, DSG3, and LCE3D (Fig. 4a–c). PAF1 also binds to the same differentiation genes as SPT6 and is required for their elongation (Fig. 4d, e). These results suggest that PAF1 and SPT6 are necessary for

differentiation by promoting the elongation of differentiation genes and validates the in vitro model of elongation in a tissue setting.

**SPT6 maintains open chromatin regions for P63 binding sites and blocks chromatin accessibility for transcription factors specifying other lineages.** Next, we wanted to determine if loss of transcription elongation on differentiation genes due to SPT6 depletion led to altered chromatin accessibility at those genes. SPT6 has been shown to promote H3K27 demethylation and thus knockdown of SPT6 could result in increased repressive chromatin (H3K27me3)[47]. To do this, CTLi and SPT6i differentiated cells were subjected to ATAC-Seq. The vast majority (83.52%) of the genome did not change in chromatin accessibility (Fig. 5a). In total, 10.74% of the genomic regions became less accessible upon SPT6 knockdown which were primarily located in intergenic (42.29%) and intronic (44.78%) regions (Fig. 5b and Supplementary Data 7a). In total, 5.73% of the genomic regions became more accessible upon SPT6 loss which were also located in intergenic (44.21%) and intronic (45.87%) regions (Fig. 5c and Supplementary Data 7b). Plotting the ATAC-Seq signal on the 1465 genes that were downregulated upon SPT6 knockdown showed that there was a slight increase of chromatin accessibility around the TSS of those genes (Fig. 5d). This further supports the idea that the genes dependent on SPT6 for expression is mediated through transcription elongation rather than a closure of chromatin. Interestingly, a transcription factor motif search at the 10.74% of the less accessible genomic regions showed P63 as the most significantly enriched factor suggesting that P63 binding sites are lost upon SPT6 depletion (Fig. 5e). Notably, the less accessible genomic regions were also enriched for FOSL2 and

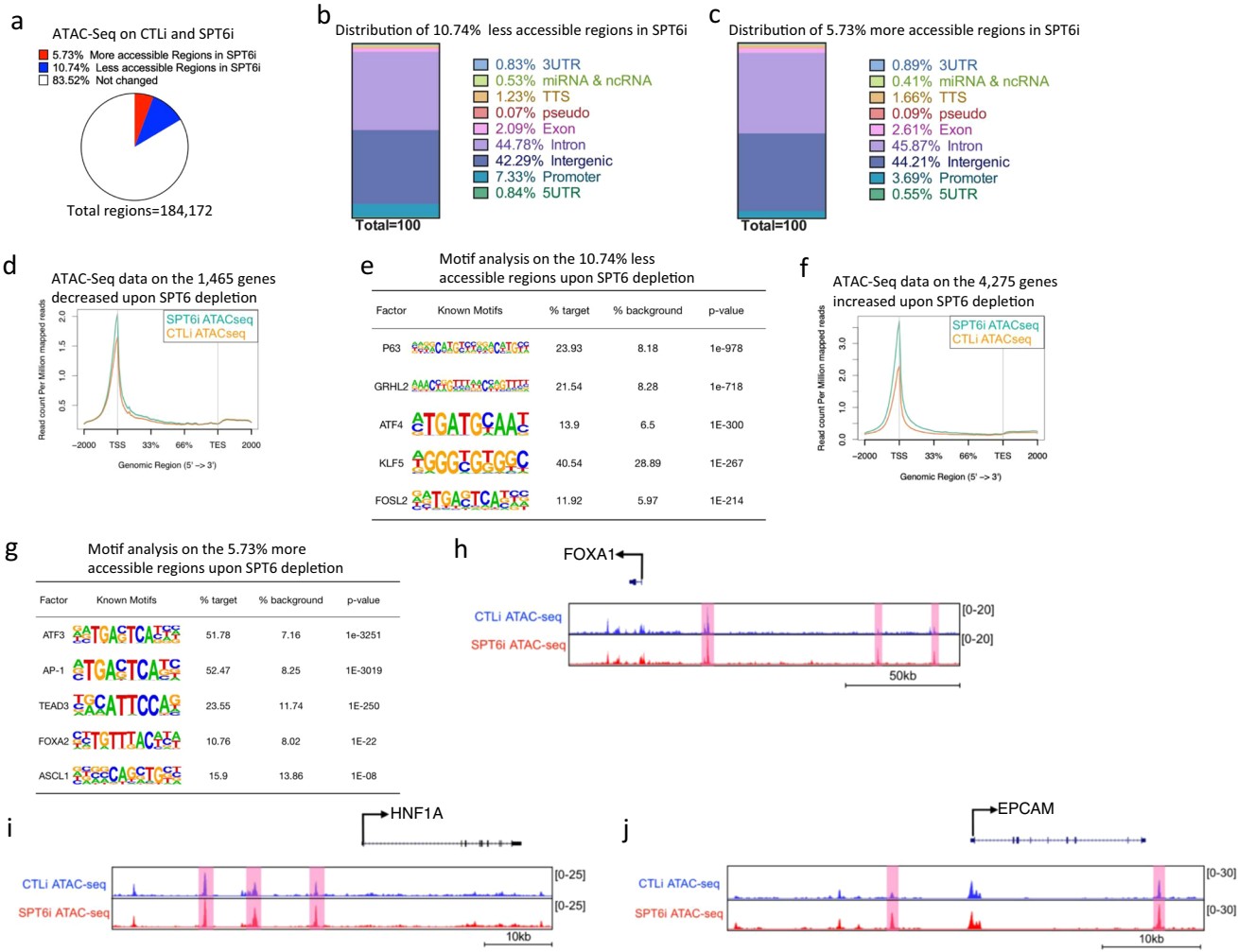

**Fig. 5 Depletion of SPT6 leads to loss of chromatin accessibility for P63 binding sites and gain of accessibility for transcription factors specifying other lineages. a** Pie chart of the percentage of regions with more or less accessible chromatin upon SPT6 depletion. CTLi and SPT6i samples were differentiated for 3 days and subjected to ATAC-Seq. $N = 2$ independent experiments. **b** Distribution of regions with less accessible chromatin upon SPT6 knockdown. **c** Distribution of regions with more accessible chromatin upon SPT6 loss. **d** ATAC-Seq signal for the 1465 genes reduced in expression upon SPT6 knockdown. ATAC-Seq signal for CTLi is shown in orange while SPT6i is shown in teal. Read count per million mapped reads is shown on the Y-axis. The X-axis represents regions across genes. TSS is transcription start site and TES is transcription end site. **e** Transcription factor motif search for the 10.74% regions with less accessible chromatin upon SPT6 knockdown. p value calculated using hypergeometric distribution. **f** ATAC-Seq signal for the 4275 genes increased in expression upon SPT6 knockdown. ATAC-Seq signal for CTLi is shown in orange while SPT6i is shown in teal. **g** Transcription factor motif search for the 5.73% regions with more accessible chromatin upon SPT6 knockdown. p value calculated using hypergeometric distribution. Gene tracks showing intestinal genes *FOXA1* (**h**), *HNF1A* (**i**), and *EPCAM* (**j**). ATAC-Seq signals for CTLi (blue) and SPT6i (red) are shown. Regions marked in pink denote increased chromatin accessibility upon SPT6 depletion. Y-axis represents reads per million. X-axis denotes regions along the gene.

KLF5 binding sites suggesting that SPT6 may directly regulate the expression of these paused regulated transcription factors that are necessary for epidermal differentiation (Fig. 5e and Supplementary Fig. 2c). Plotting of the ATAC-Seq data on the 4275 genes that were upregulated upon SPT6 knockdown showed a large increase in chromatin accessibility at the TSS of those genes (Fig. 5f). A transcription factor motif search at the 5.73% of the more accessible genomic regions showed enrichment for factors such as ASCL1 and FOXA2 which are lineage-specific transcription factors that regulate neuronal and intestinal development (Fig. 5g). A look at transcription factors and structural proteins important for endoderm/intestinal fate such as FOXA1, HNF1A, EPCAM showed increased chromatin accessibility upon SPT6 depletion at regions proximal to these genes (Fig. 5h–j). These results suggest that SPT6 may potentially be regulating the transcription elongation of P63 and the other pause regulated transcription factors such as FOSL2 and KLF5. Thus, without

SPT6, P63, and the other epidermal transcription factor gene expression may go down which would then lead to closure of their bound sites. The increase in chromatin accessibility at non-epidermal lineages may be a secondary effect of SPT6 regulation. SPT6 may directly regulate factors such as P63 and these transcription factors may in turn regulate non-epidermal genes.

**SPT6 suppresses an intestinal fate by promoting the expression of P63**. Since there was increased chromatin accessibility near genes important for specifying non-epidermal lineages, we asked if SPT6 knockdown triggers a different tissue fate. To test this, genes upregulated ≥10 fold (472 genes) upon SPT6 depletion were used to query the Human Gene Atlas and ARCHS4 Tissues database through Enrichr to determine which tissue or cell type the 472 genes correlated with the most[44,48,49]. The Human Gene Atlas contains RNA-Seq samples from 37 different types of

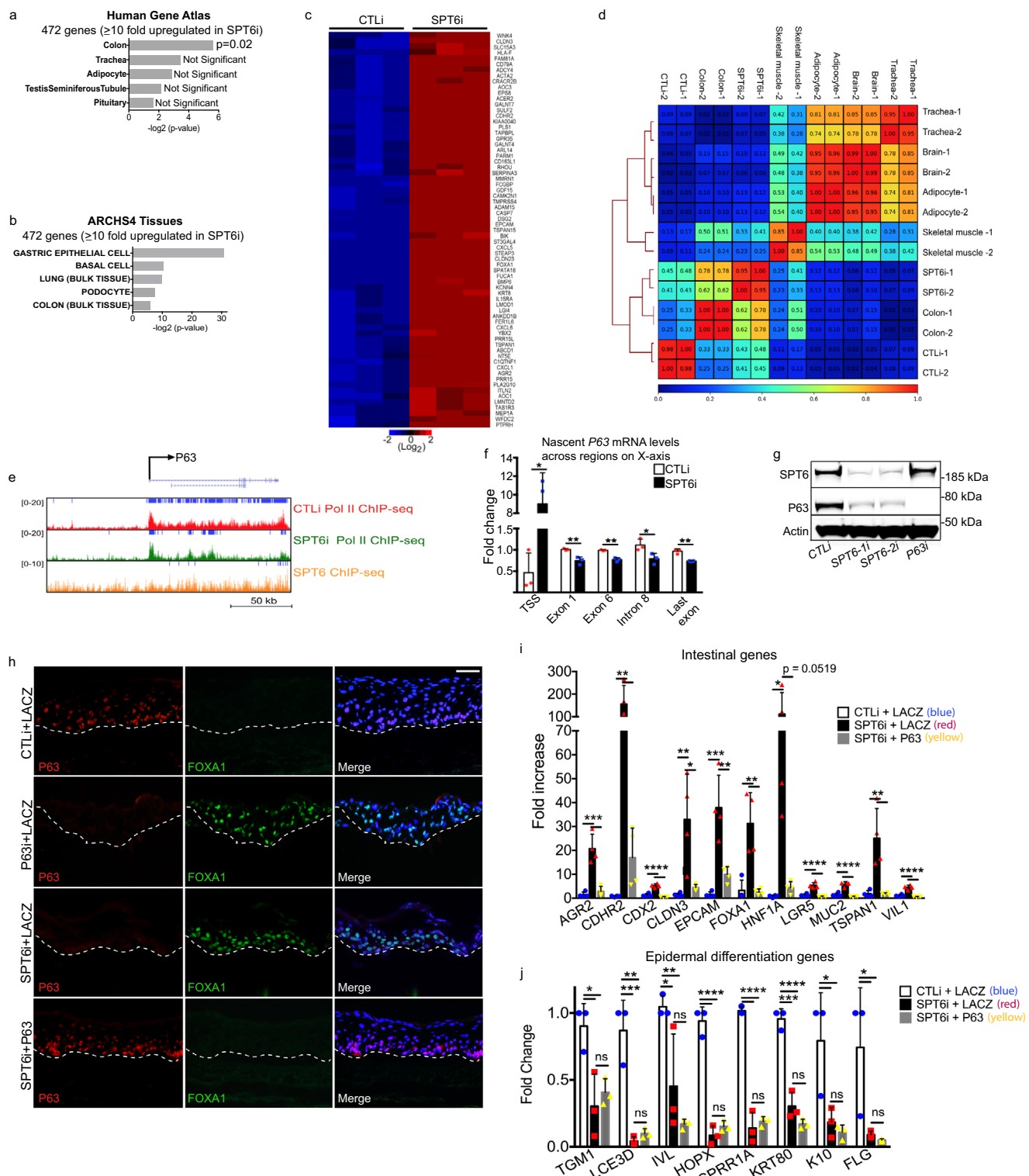

normal tissue and ARCHS4 is a compilation of 84,863 human RNA-Seq datasets covering a broad range of cell types and tissues. The only significant tissue that matched with the 472 upregulated genes was colon from the Human Gene Atlas (Fig. 6a). Similarly, gastric epithelial cells were the top match from the ARCHS4 Tissues database and matches to colon was also significant in the same database (Fig. 6b). A comparison between CTLi and SPT6i tissue showed that these intestinal genes (identified from the ARCHS4 database) were highly upregulated upon SPT6 loss and includes genes such as *HNF1A*, *MUC2*, *LGR5*, *VIL1*, *FOXA1*,

*CDX2*, *CLDN3*, *AGR2*, *TSPAN1*, *EPCAM*, and *CDHR2* (Fig. 6c and Supplementary Fig. 4a). We also wanted to determine if the entire SPT6 gene expression signature matched with that of colon. We performed Pearson correlations of the SPT6 knock-down RNA-Seq with that of previously published RNA-Seq data derived from human brain, skeletal muscle, adipocyte, trachea, and colon[50–52]. The SPT6 signature was most similar to colon with an average Pearson coefficient of 0.7 followed by skeletal muscle (0.305) and low correlations with brain (0.085), adipocyte (0.125), and trachea (0.07) (Fig. 6d). The control human

**Fig. 6 Depletion of SPT6 leads to an intestinal-like phenotype due to loss of P63 expression. a** The top 472 genes upregulated upon SPT6 depletion were searched in the Human Gene Atlas database to determine tissues with most similar gene expression using hypergeometric distribution. **b** The top 472 genes upregulated upon SPT6 depletion were used in the ARCHS4 database to determine tissues or cell types with most similar gene expression using hypergeometric distribution. **c** Heatmap of the intestinal genes (from the ARCHS4 database) upregulated upon SPT6 knockdown. Heatmap is shown in Log2. **d** Pearson correlation and clustering of CTLi and SPT6i gene expression signatures with brain, colon, adipocyte, trachea, and skeletal muscle. Replicate RNA-Seq samples are shown for each tissue. **e** Gene track of *P63*. SPT6 ChIP-Seq is shown in orange. CTLi (red) and SPT6i (green) Pol II ChIP-Seq are also shown. *Y*-axis shows reads per million and blue bars over gene track represent significant peaks. **f** Measurement of nascent *P63* mRNA levels in CTLi and SPT6i cells differentiated for 3 days. Cells were pulsed with 5-ethynyl uridine for 4 h and nascent transcripts purified. Levels of nascent *P63* mRNA levels were measured using RT-QPCR at the regions shown on the *X*-axis. $N = 3$ independent experiments, mean values are shown with error bars = SD, two-sided *t*-test. * = 0.012 for TSS, ** = 0.0074 for exon 1, ** = 0.0012 for exon 6, * = 0.044 for intron 8, ** = 0.0024 for last exon. **g** Western blot for SPT6 and P63 protein levels in CTLi, SPT6-1i, SPT6-2i, and P63i cells. Representative image is shown, $n = 3$ independent experiments. **h** Immunofluorescent staining of P63 (red) and FOXA1 (green) in day 6 regenerated human epidermis. Samples include CTLi + LACZ, P63i + LACZ, SPT6i + LACZ, and SPT6i + P63. LACZ and P63 are exogenous retroviral expression of the open reading frames of each gene. Merged image includes Hoechst staining of nuclei. $n = 3$ independent experiments, scale bar = 60 μm. **i** RT-qPCR quantifying the relative mRNA expression levels of intestinal genes in regenerated human epidermis (day 6). Samples include CTLi + LACZ, SPT6i + LACZ, and SPT6i + P63. LACZ and P63 are exogenous retroviral expression of the open reading frames of each gene. qPCR values were normalized to *L32*. N = 4 independent experiments. $p = 0.0008$ for AGR2, $p = 0.0089$ for CDHR2, $p = 0.0002$ for CDX2, $p = 0.0165$ for CLDN3, $p = 0.0018$ for EPCAM, $p = 0.0062$ for FOXA1, $p = 0.0006$ for LGR5, $p = 0.0004$ for MUC2, $p = 0.0099$ for TSPAN1, $p = 0.0004$ for VIL1. **j** RT-qPCR quantifying the relative mRNA expression levels of epidermal differentiation genes in regenerated human epidermis (day 6). Samples include CTLi + LACZ, SPT6i + LACZ, and SPT6i + P63. LACZ and P63 are exogenous retroviral expression of the open reading frames of each gene. qPCR values were normalized to *L32*. N = 3 independent experiments. $p = 0.0241$ for TGM1, $p = 0.0033$ for LCE3D, $p = 0.0003$ for HOPX, $p = 0.0002$ for SPRR1A, $p = 0.0009$ for KRT80, $p = 0.0478$ for K10. ns not significant. Mean values are shown with error bars = SD. *$p < 0.05$, **$p < 0.01$, ***$p < 0.001$, ****$p < 0.0001$ (two-sided *t*-test for **i**, **j**).

---

regenerated skin samples showed low correlations (0.29) with colon and SPT6 knockdown samples (0.44) (Fig. 6d). This suggests that loss of SPT6 caused the tissue to resemble more of a colon than skin phenotype. Next, we wanted to elucidate the mechanism behind this altered cell fate. P63 is a master regulator of epidermal fate with p63 knockout mouse embryos developing intestine-like metaplasia[53]. SPT6 bound to *P63* and is required for its transcription elongation and mRNA expression (Fig. 6e and Supplementary Fig. 3d and Supplementary Data 4). Measurement of nascent *P63* mRNA by pulsing control and SPT6 depleted cells with 5-ethynyl uridine also showed a significant decrease of *P63* mRNA from the gene body (exon 1, exon 6, intron 8, and the last exon) in SPT6 knockdown cells (Fig. 6f). Loss of *P63* mRNA from the gene body was accompanied by a large increase of the message accumulating near the TSS region suggesting that SPT6 is necessary for elongation of *P63* transcripts (Fig. 6f). This resulted in loss of P63 on the protein level using in vitro Ca2+ induced differentiation and regenerated human tissue (Fig. 6g and Supplementary Fig. 4b). Intersection of the upregulated genes due to SPT6 knockdown (RNA-Seq) with genes bound by P63 [(P63 ChIP-Seq in differentiated primary human keratinocytes[54]] showed that 45.9% of the genes overlapped (Supplementary Fig. 4c). This suggests that p63 may act as a repressor on those genes. Supporting this, the chromatin accessibility increased at P63 bound intestinal genes such as *HNF1A*, *FOXA1*, *CDHR2*, and *EPCAM* upon SPT6 loss (Supplementary Fig. 4d–g). Furthermore, knockdown of SPT6 caused decreased P63 binding at those genes (Supplementary Fig. 4h). Loss of P63 in regenerated human tissue also resulted in the upregulation of the same set of intestinal genes as SPT6i including *HNF1A*, *MUC2*, *LGR5*, *VIL1*, *FOXA1*, *CDX2*, *CLDN3*, *AGR2*, *TSPAN1*, *EPCAM*, and *CDHR2* (Supplementary Fig. 4i). This suggests that SPT6 may be promoting the expression of P63 to suppress the intestinal genes. If this were true, then ectopic expression of P63 should be able to block induction of intestinal gene expression in SPT6 knockdown tissue. Control epidermis, generated from keratinocytes treated with control siRNAs and retroviral expression of LACZ (CTLi + LACZ), had robust P63 and no detectable FOXA1 expression (Fig. 6h). FOXA1 has been shown to be one of three transcription factors necessary to reprogram murine embryonic fibroblasts into

an intestinal fate[55]. Expression of P63 was lost in SPT6 (SPT6i + LACZ) or P63 (P63i + LACZ) knockdown tissue which was replaced by robust expression of FOXA1 (Fig. 6h). Importantly, retroviral expression of P63 suppressed FOXA1 expression as well as *HNF1A*, *MUC2*, *LGR5*, *VIL1*, *FOXA1*, *CDX2*, *CLDN3*, *AGR2*, *TSPAN1*, *EPCAM*, and *CDHR2* mRNA expression in SPT6 depleted tissue but did not restore epidermal differentiation gene expression (Fig. 6h–j). These results suggest that SPT6 directly regulates epidermal differentiation genes through elongation and indirectly prevents an intestinal fate by promoting the expression of P63 and possibly other paused transcription factors.

To explore whether loss of SPT6 can lead to formation of intestinal epithelial structures, we cultured control and SPT6 knockdown keratinocytes as single cells in matrigel to form 3D organoids. The cells were grown in proliferation (promotes intestinal stem cell expansion) or differentiation media (promotes budding into crypt-like structures via differentiation of stem cells into mature crypt cells) in the presence of Wnt3a, R-spondin, and noggin. These culture conditions have previously been shown to promote the growth of intestinal stem cells into large spherical structures containing a polarized epithelium around a centered lumen with budding crypt-like domains[55–58]. In the case of control cells, ~90% of the colonies grew as large solid spheroids (without a lumen) regardless of whether they were grown in proliferation or differentiation medium (Fig. 7a, e, f). In contrast, the proportion of solid spheres was decreased to ~60% in proliferation and ~45% in differentiation media in SPT6 depleted cells (Fig. 7e, f). This drop in solid spheres was replaced by a concomitant increase in spheroid with lumen (single or multiple: Fig. 7c, right panel), budding (Fig. 7b, g; ~8% in proliferation media and ~15% in differentiation media), and apoptotic events (Fig. 7c, left panel). These phenotypes mirrored that seen in healthy human colon organoids derived from biopsies obtained during routine colonoscopy (Fig. 7d). Notably, no budding structures were detected in control cells (Fig. 7g). SPT6 depleted organoids also expressed intestinal genes such as *HNF1A*, *FOXA1*, *LGR5*, *KRT20*, *MUC2*, and *CDHR2* (Fig. 7h). These data suggest that SPT6 loss leads to a morphologic and gene expression transformation that resembles an intestinal fate.

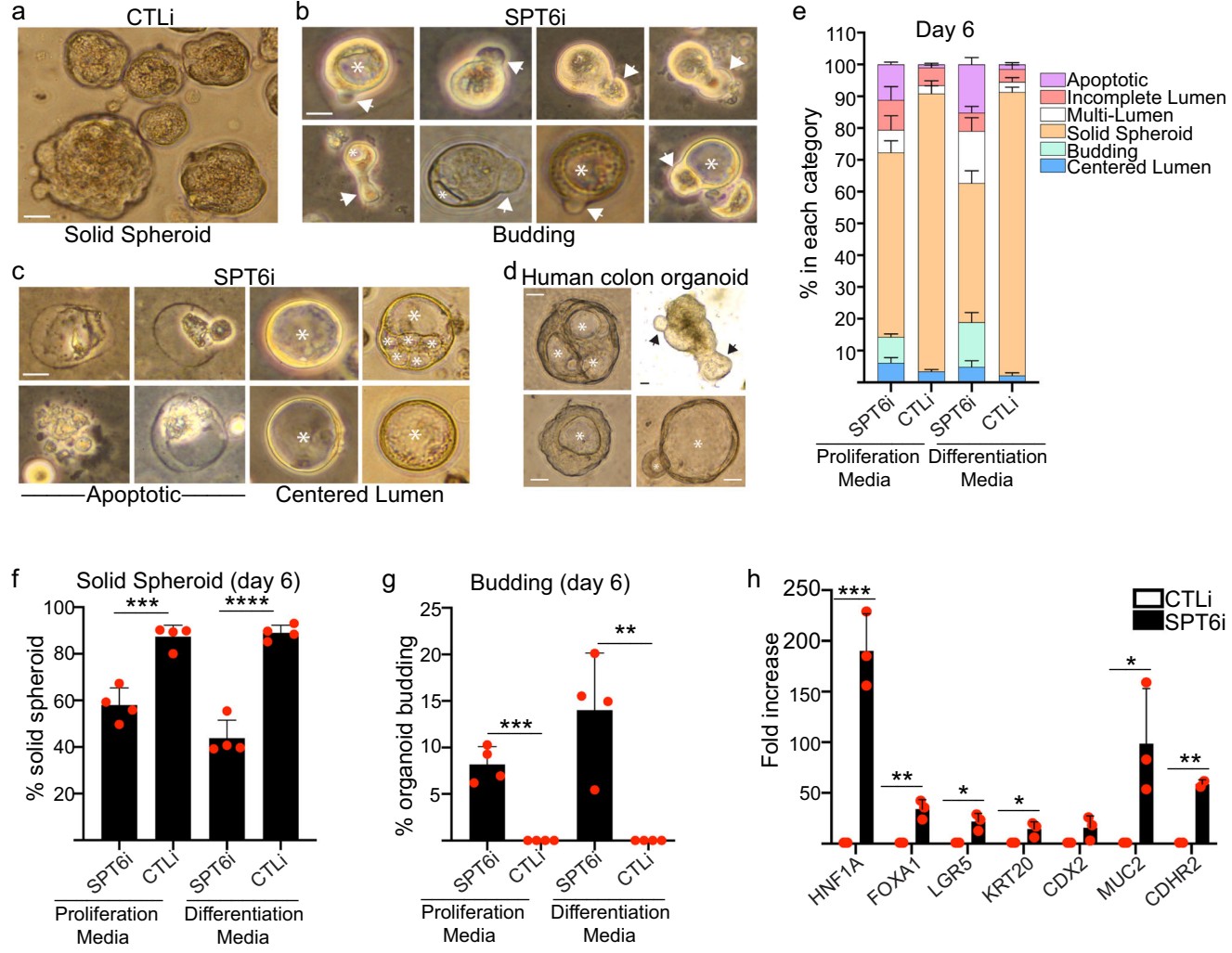

**Fig. 7 SPT6 knockdown cells can generate structures resembling intestinal organoids. a** Representative image of control (CTLi) knockdown primary human keratinocytes grown in matrigel in the presence of Wnt3a, R-spondin, and noggin. **b, c** Images of SPT6 (SPT6i) knockdown primary human keratinocytes grown in matrigel in the presence of Wnt3a, R-spondin, and noggin. A montage of images of budding organoids is shown in **b**. A montage of images of apoptotic (**c**: left panel) and lumen-containing spheroids (**c**: right panel) is shown. **d** Representative images of healthy human colon organoids are shown. Arrows denote budding and asterisks denote centered lumen for **b–d**. Scale bar = 100 μm for **a–d**. N = 4 independent experiments which all showed similar results. **e** Stacked bar graphs display the percent (%) of SPT6 (SPT6i) or control (CTLi) knockdown organoids that are budding, apoptotic, with incomplete lumen, multi-lumen, solid spheroids, or centered lumen. N = 4 independent experiments. **f** Bar graphs display the percent (%) of SPT6i or CTLi solid spheroids. N = 4 independent experiments. *** = 0.0006 for proliferation media. **g** Bar graphs display the percent (%) of SPT6i or CTLi organoids that are budding. N = 4 independent experiments. *** = 0.0001 for proliferation media, ** = 0.0039 for differentiation media. **h** RT-qPCR quantifying the relative mRNA expression levels of intestinal genes in CTLi and SPT6i organoids. qPCR values were normalized to *L32*. *** = 0.0009 for HNF1A, ** = 0.0037 for FOXA1, * = 0.012 for LGR5, * = 0.0332 for KRT20, * = 0.036 for MUC2. N = 3 independent experiments. Mean values are shown with error bars = SD. *$p < 0.05$, **$p < 0.01$, ***$p < 0.001$, ****$p < 0.0001$ (two-sided *t*-test, **f–h**).

## Discussion

Promoter proximal pausing has been prominently studied in *Drosophila* embryos and ESCs in the context of regulation of developmental genes[21–24]. Pol II may be paused on developmental genes to poise critical genes for rapid induction upon the appropriate signal during development[59]. Surprisingly, there is not much known about this process in adult stem and progenitor cell fate decisions. Here, we have shown that ~30% of induced epidermal differentiation genes already have paused Pol II binding in stem and progenitor cells. Pol II accumulated at the promoter proximal regions. In total, 155 of these genes were categorized as highly paused with a TR ≤ 0.25, while 539 genes were moderately paused. Upon induction of differentiation, there was a dramatic decrease in Pol II accumulation/pausing at the

promoter proximal regions in the 694 differentiation genes which also resulted in a significant increase in their TRs. These important genes include *DSG3* and *ABCA12* where loss of function results in skin diseases such as pemphigus vulgaris and harlequin ichthyosis[60,61]. Notably, the paused genes were also enriched for transcription factors critical for the differentiation process such as *OVOL1, GRHL1, KLF4, KLF5, MAF, FOSL2*, and *CEBPB*[9,10,13,15,16,62,63]. This suggests that these pause regulated transcription factors may recruit Pol II to turn on the non-pause regulated differentiation genes.

If transcriptional elongation regulates ~30% of epidermal differentiation genes, then factors that regulate pause release/transcription elongation should be critical for the differentiation process. Through an RNAi screen of these factors, we identified

SPT6 to be a critical regulator of epidermal differentiation. It is interesting to note that none of the other factors had any impacts on differentiation. This is in line with observations in ESCs where specific elongation/pause release factors govern distinct ESC behavior. For example, Paf1C and Phf5a are important for the elongation of genes coding for pluripotency transcription factors, while Ell3 is necessary for establishing Pol II at developmental genes in ESCs[64–66]. In addition, NELF is necessary for regulating cell cycle genes in ESCs suggesting that different elongation/pause release factors have different targets as well as functions even in the same cell type[67]. Thus, it is possible that the factors we screened may have other functions in the skin such as in promoting self-renewal. This will be an interesting area for future investigation.

Loss of SPT6 blocked early (KRT1, KRT10) and late (FLG, LOR) differentiation as well as stratification of the epidermis. Global gene expression profiling of SPT6 knockdown tissue showed that 1465 genes were downregulated that had GO terms related to skin development and differentiation. ChIP-Seq of SPT6 showed that SPT6 bound to 6307 genes that were enriched in epidermis development and keratinocyte differentiation GO terms. However, it should be noted that SPT6 does not exclusively bind differentiation genes as it also binds and regulates the expression of basal layer specific genes such as *KRT5, P63,* and *KRT14* (Supplementary Data 4 and 5). It is currently unclear how SPT6 binds with such specificity to epidermal genes. It is possible that SPT6 is recruited by an early epidermal differentiation transcription factor to promote elongation. Growth promoting transcription factors such as YAP and C-MYC have been found to promote Pol II pause release instead of Pol II recruitment to target genes, suggesting that a transcription factor may potentially recruit SPT6 to promote transcriptional elongation of epidermal genes[21,68]. SPT6 bound to 34.5% of upregulated differentiation genes and nearly 40% of the genes with decreased expression upon SPT6 depletion supporting the idea that SPT6 is important for epidermal gene expression. SPT6 binding to these genes resembled the profile of Pol II suggesting it may act as an elongation factor. To demonstrate this, Pol II ChIP-Seq was used to determine the binding profile of Pol II in the absence of SPT6. On SPT6 bound genes, loss of SPT6 increased promoter proximal pausing of Pol II and reduced Pol II binding throughout the gene suggesting SPT6 promotes transcription elongation. Pol II binding to the gene bodies of the 694 paused differentiation genes was also decreased that correlated with a significant decrease in their TRs. In contrast, SPT6 knockdown had minimal impact on the RNA Pol II binding profile on non-SPT6 bound genes. Our results suggest that SPT6 binds to and is necessary for the transcriptional elongation of epidermal genes. The current model of transcription elongation elucidated by cryo-EM suggests that the complex of pTEFb, SPT6, and PAF is required for transcription elongation activation in vitro[30]. However, it is unclear whether this same mode of regulation occurs in a tissue setting. We have shown here that PAF1 and SPT6 work in the same genetic pathway where they are both required for epidermal differentiation. Additionally, PAF1 binds to the same genes as SPT6 and loss of PAF1 results in Pol II stalling at the TSS regions of differentiation genes. Our data suggest that SPT6 promotes elongation in conjunction with PAF1 to promote differentiation in a tissue setting that is in line with the in vitro model of elongation. It will be interesting to determine whether this same group of proteins is necessary for elongation in other organs of the body. This is important in light of our small RNAi screen where other major known elongation factors had no apparent impact on epidermal differentiation. This suggests that the elongation factors that have been primarily characterized in vitro and thought to work together in a complex may have differing and even opposite functions in tissue.

The next question we asked was whether a loss of transcription elongation impacted chromatin accessibility through ATAC-Seq. The vast majority of the genome had no change in accessibility upon SPT6 depletion whereas 5.73% gained, while 10.74% lost accessibility. Interestingly, there was a slight increase in the chromatin accessibility of the 1465 genes decreased upon SPT6 loss suggesting that loss of transcription elongation does not lead to a loss of chromatin accessibility. An examination of the 4275 genes upregulated upon SPT6 depletion showed an even greater increase in chromatin accessibility in their TSS. Moreover, a motif search of the regions that gained chromatin accessibility was surprisingly enriched for transcription factors motifs that specify other lineages such as the brain and gut. This suggested the possibility that SPT6 loss resulted in an altered cell fate. To test this, we took the 472 genes that are ≥10 fold upregulated in expression upon SPT6 depletion and found that they were enriched for colon and gastric epithelial cell signatures. In addition, pearson correlations of the entire gene expression signature of SPT6i cells showed high correlation (avg. 0.7) with RNA-Seq signatures derived from human colon but not skeletal muscle or brain. Since SPT6 promotes transcriptional elongation of epidermal genes, how does its loss lead to an intestinal-like phenotype. A motif search of the regions that lost chromatin accessibility showed the transcription factor P63 as the most significant. P63 is a transcription factor necessary for epidermal fate determination and part of SPT6's loss of function phenotype may be mediated through this factor. Supporting this, SPT6 knockdown resulted in Pol II depletion from the gene body of P63 which led to loss of P63 expression on the mRNA and protein level. Knockdown of P63 also resulted in upregulation of intestinal genes such as *FOXA1, CDX2, VIL1, CLDN3, AGR2, EPCAM, LGR5,* and *CDHR2*. These results are consistent with observations that p63 null mouse embryos develop intestine-like metaplasia[53]. Another group also showed that in embryonic epithelia, p63 knockout prevented stratification which resulted in a single layer epithelium with expression of mesodermal/muscle genes[69]. More recently, Qu et al. used keratinocytes derived from EEC patients to show that epidermal cell identity was lost in cells with p63 DNA binding mutations[70]. These mutant cells also had upregulated expression of neuronal and mesenchymal genes. It is notable that SPT6 loss had the highest Pearson correlation with colon and followed by skeletal muscle suggesting regulation of P63 may be partially responsible for some of its phenotype. In support of this, over 45% of the genes upregulated upon SPT6 knockdown had P63 binding suggesting that P63 may be acting as a repressor on these genes. SPT6 knockdown also resulted in increased chromatin accessibility at P63 bound intestinal genes including *HNF1A, FOXA1, EPCAM,* and *CDHR2*. Furthermore, ectopic expression of P63 prevented the induction of intestinal genes but did not restore epidermal differentiation gene expression in SPT6 depleted tissue. It is also possible that other pause regulated transcription factors contribute to the intestinal-like phenotype since SPT6 knockdown also causes loss of their expression. These other transcription factors may also contribute to epidermal identity as well as suppressing other cell fates.

We have also shown here that SPT6 knockdown cells grown in matrigel in the presence of Wnt3a, R-spondin, and noggin can form structures that resemble intestinal organoids with the presence of centered lumen and rudimentary budding structures. However, it is unclear how similar and whether these cells behave like real intestinal organoids. It will be interesting in the future to determine whether organoids derived from SPT6 depleted human keratinocytes can rescue mice in colon injury models.

Here, we have found a prominent role for SPT6 in promoting epidermal differentiation through elongation. SPT6 directly promotes the elongation of structural differentiation genes as well as

transcription factors necessary to turn on the rest of the differentiation program (differentiation genes without Pol II loading in undifferentiated cells). Lastly, SPT6 indirectly suppresses an intestinal cell fate through master regulators of epidermal fate such as P63. In summary, we have elucidated a critical pathway where regulation of promoter pausing/transcriptional elongation through SPT6 plays a major role in somatic cell fate decisions.

## Methods

**Primers and siRNA sequences**. All primers and siRNA sequences can be found in Supplementary Data 8.

**Primary cells**. Primary human epidermal keratinocytes were derived from neonatal foreskin and cultured in EpiLife medium (ThermoFisher: MEPI500CA) supplemented with human keratinocyte growth supplement (ThermoFisher: S1001K) and pen/strep. Proliferating, non-differentiated keratinocytes were cultured in subconfluent conditions. To induce epidermal differentiation, primary human keratinocytes were plated at full confluence in the presence of 1.2 mM calcium for 3 days. Phoenix cells were cultured in DMEM with 10% fetal calf serum.

**Organotypic cultures of human epidermis**. For the organotypic skin cultures, one million control or SPT6 knockdown cells were seeded onto devitalized human dermis to regenerate human epidermis[45,71]. Human dermis was purchased from the New York Firefighters Skin Bank. Dermis seeded cells were raised to the air liquid interface to promote differentiation and stratification. Tissue was harvested 6 days after initial seeding. Half of each sample was embedded into OCT for sectioning and the other half was collected for RNA extraction.

**siRNA transfection**. siRNAs targeting human SPT6 (final concentration 10 nM) or control siRNAs were transfected into keratinocytes using Lipofectamine RNAiMAX (Thermo: 13778-500) reagent according to the manual and incubated for 18 h.

**RNA isolation and RT-QPCR**. Total RNA from cells or tissue was extracted using the GeneJET RNA purification kit (Thermo Scientific: K0732) and quantified using a Nanodrop. In total, 1 μg of total RNA was reversed transcribed using the Maxima cDNA synthesis kit. Quantitative PCR was performed using the BioRad LFX96 real-time system. L32 was used as internal control for normalization.

**Western blotting**. Twenty microgram of the cell lysates were used for immunoblotting and resolved on 10% SDS-PAGE and transferred to PVDF membranes. Primary antibodies used include beta-actin (Santa Cruz: Sc-47778) at 1:5000, ELL (Cell Signaling Technology: 14468) at 1:500, SPT5 (Bethyl: A300-869A) at 1:500, p63 (Abcam: ab124762) at 1:1000, and SPT6 (Bethyl: A300-801A) at 1:1000. Secondary antibodies including IRDye 800CW (LI-COR: 926-32212) donkey anti-mouse and IRDye 680RD (LI-COR: 926-68073) donkey anti-rabbit were used at 1:10000.

**Histology and immunofluorescence**. Cultured cells or tissue were fixed in 4% paraformaldehyde for 11 min followed by blocking in PBS with 2.5% normal goat serum, 0.3% triton X-100, and 2% bovine serum albumin for 30 min Primary antibodies used were P63 (Rabbit, Abcam: ab124762) at 1:1000, P63 (Mouse, Abcam: ab735) at 1:100, FOXA1 (Cell Signaling: 53528S) at 1:1000, Loricrin (Abcam: Ab198994) at 1:1000, Filaggrin (Abcam: Ab3137) at 1:200, MKi67 (Abcam: Ab16667) at 1:300, Keratin 10 (Abcam: Ab9025) at 1:500, Keratin 1 (Biolegend: 905204) at 1:500 for 1 h. The secondary antibodies used were Alexa 555 conjugated goat anti-mouse IgG (Thermo: A11029) or Alexa 488 conjugated donkey anti-rabbit IgG (Thermo: A21206) both at 1:500. Nuclear dye, Hoechst 33342 (Thermo:H3570) was used at 1:1000.

**RNA-sequencing (RNA-seq) and library preparation**. One million control or SPT6i cells were placed on devitalized human dermis and harvested 6 days later. Three biological triplicates were obtained for both CTLi and SPT6i and total RNA was isolated using the GeneJET RNA (Thermo: K0732) purification kit and quantified by Nanodrop. RNA-seq was performed using the Illumina HiSeq 4000 machine at the Institute of Genomic Medicine core facility at UCSD. RNA-seq libraries were prepared with TruSeq RNA Library Prep Kit then multiplexed and ~40 million reads per sample were obtained.

**Chromatin immunoprecipitation sequencing (ChIP-seq), library preparation, and ChIP-QPCR**. Ten million cells and 5 μg of antibody were used for each antibody pulldown experiment for ChIP[46,72,73]. ChIP was performed using the following antibodies: SPT6 (Bethyl: A300-801A), RNA Pol II (Active Motif: 91151), PAF1 (Bethyl: A300-172A), p63 (Abcam: Ab735), Rabbit IgG (Millipore: 12-370),

and mouse IgG (Abcam: Ab18413). Cells for the RNA Pol II ChIP-QPCR or ChIP-Seq were fixed at a final concentration of 1% formaldehyde. Cells for the SPT6 ChIP-QPCR or ChIP-Seq were fixed in both formaldehyde (1% final concentration, Thermo: 28908) and disuccinimidyl glutarate (2 mM final concentration, Thermo: 20593). QPCR results are represented as a percentage of input DNA.

For ChIP-Seq, the ChIP DNA library was prepared using the TruSeq DNA sample prep kit (Illumina). Sequencing was done on the HiSeq 4000 System (Illumina) using single 1 × 75 reads at the Institute for Genomic Medicine Core, UCSD.

**ATAC-sequencing (ATAC-seq) and library preparation**. One million keratinocytes were transfected with control siRNA or SPT6 siRNA 1 day before seeded on a 12-well plate (Wuxi, CELL NEST, China, Cat# 712001). After inducing differentiation for 3 days, the cells were washed briefly with phosphate-buffered saline (PBS) two times. Permeabilized nuclei were obtained by resuspending cells in 250 μl Nuclear Permeabilization Buffer [(10 mM Tris-HCl (pH 7.4), 10 mM NaCl, 3 mM MgCl₂, 0.1% Tween-20, 0.1% IGEPAL-CA630, 0.01% digitonin in Molecular biology water], and incubating for 5 min on a rotator at 4 °C. Nuclei were then pelleted by centrifugation for 5 min at $500 \times g$ at 4 °C. The pellet was resuspended in 25 μl ice-cold Tagmentation Buffer [33 mM Tris-acetate (pH = 7.8), 66 mM K-acetate, 11 mM Mg-acetate, 16% DMF in Molecular biology water]. An aliquot was then taken and counted by hemocytometer to determine nuclei concentration. Approximately 50,000 nuclei were resuspended in 10 μl ice-cold Tagmentation Buffer, and incubated with 1 μl Tagmentation enzyme at 37 °C for 30 min with shaking 500 rpm. The tagmented DNA was purified using MinElute PCR purification kit.

The libraries were amplified using NEBNext High-Fidelity 2X PCR Master Mix with primer extension at 72 °C for 5 min, denaturation at 98 °C for 30 s, followed by eight cycles of denaturation at 98 °C for 10 s, annealing at 63 °C for 30 s, and extension at 72 °C for 60 s. Amplified libraries were then purified using MinElute PCR purification kit, and two size selection steps were performed using SPRIselect bead at 0.55X and 1.5X bead-to-sample volume rations, respectively. Sequencing was performed at the UC San Diego IGM Genomics Center on an Illumina NovaSeq 6000 using the 100 bp paired-end protocol. The ATAC-seq was performed in CTLi and SPT6i cells in duplicates.

**Nascent P63 mRNA measurement**. Control and SPT6 knockdown cells were differentiated for 3 days and pulsed with 5-ethynyl uridine (EU) for 4 h. After the pulse, nascent RNA labeled with EU is used in a copper catalyzed click reaction with azide-modified biotin. The EU-biotin labeled RNA is then purified and selectively isolated using streptavidin magnetic beads according to manufacturer's protocol (Click-iT Nascent RNA Capture Kit: Invitrogen C10365). RT-QPCR was used to measure levels of nascent P63 mRNA.

**Intestinal organoid culture**. Control and SPT6 knockdown keratinocytes were cultured under intestinal organoid conditions[56]. Cells were seeded in Matrigel (Corning, 354234) domes at $5 \times 10^4$ cells per well in a 24-well plate (USA Scientific, CC7682-7524). The plate was inverted and incubated at 37 °C to allow for Matrigel polymerization. After 10 min, 500 ul of proliferation media (50% conditioned media prepared from L-WRN cells (ATCC, CRL-3276) containing Wnt3a, R-spondin, and noggin) or differentiation media (5% conditioned media) was added to each well[57]. Cells were maintained in a humidified 37 °C/5% CO₂ incubator for 9–10 days. Media changes were performed every 2–3 days. Cells were monitored under a light microscope for evidence of organoid formation and crypt budding. Every 3 days, organoid lumen structures (centered, solid, multi, incomplete, budding, and apoptotic) were imaged and counted. The colon organoids were isolated from the colonic tissue specimens of healthy adult human subjects undergoing routine colonoscopy[56,74]. The isolation and biobanking of organoids from these biopsies was carried out using an approved human research protocol (IRB# 190105: PI Ghosh and Das) that covers human subject research at the UC San Diego HUMANOID Center of Research Excellence.

## Quantification and statistical analysis

**RNA-seq data processing**. Reads were aligned to the GENCODE v19 transcriptome hg19 using TopHat2 with default settings[75]. Differential expression among samples was calculated using ANOVA from the Partek Genomic Suite (Partek Incorporated). Analysis of the read count distribution indicated that a threshold of ten reads per gene generally separated expressed from unexpressed genes, so all genes with fewer than ten reads were excluded from ANOVA analysis. Gene lists for significantly upregulated or downregulated genes were created using FDR < 0.05 and ≥2-fold change. Enriched GO terms for RNA-seq differentially expressed gene sets were identified using Enrichr[44,76]. Heatmaps for the RNA-seq data were generated using Partek's Genomic Suite (http://www.partek.com/partek-genomics-suite/).

**ChIP-seq data processing**. The ChIP-seq reads were processed by the ENCODE Transcription Factor and Histone ChIP-Seq processing pipeline (https://github.com/ENCODE-DCC/chip-seq-pipeline2) on our local workstation. The reads were

first trimmed based on quality score before alignment to reference hg19; Upon alignment and deduplication, the peak calling was then carried out by MACS 2.2.4 with a cutoff $q$ value of 0.05[77,78]. The heatmaps for the ChIP-Seq data were generated using ngs.plot[79]. Gene tracks were visualized using UCSC genome browser along with annotation tracks.

**ATAC-seq data processing**. We deployed the ENCODE ATAC-seq pipeline ATAC-seq pipeline v1.7.0 (https://github.com/ENCODE-DCC/atac-seq-pipeline) on our local workstation for ATAC-seq peak calling. The raw reads were trimmed with cutadapt2.5 before aligned to the GRCh37 (hg19) using bowtie2.3[80]. The alignment files were then deduplicated using picard2.20 prior to being converted to tag files and subjected to peaking calling using MACS 2.2.4. The overlapped peaks from different samples were then merged into a consensus set of peaks prior to being divided into evenly spaced 500 bp chromatin regions over that the aligned reads could be recounted for differential enrichment analysis using DiffBind2.14[81]. DEseq2 was invoked to perform the differential analysis for accessible peaks[82]. FDR of less or equal to 0.05 was used to correct for the multiple tests.

**Calculation of travel ratio from RNA Pol II ChIP-Seq data**. To compute the TR for each gene, the following steps were implemented: (1) we defined the TSS region as 50 bp upstream to 300 bp downstream of the TSS, and the gene body as 300 bp downstream of the TSS to 1000 bp downstream of the TES; (2) the average enrichment score over a genomic region was computed using UCSC genome browser utility *bigWigAverageOverBed* with the Pol II ChIP-Seq fold change signal track; (3) the TR was then calculated as the ratio between the average enrichment score over the gene body region and the score over TSS region. As gene with multiple transcripts would bring about multiple options in defining the corresponding genomic regions, we opted to use the transcript with the highest average signal over the entire gene in the ChIP-seq control sample (input) for simplicity. The TSS reference was taken from UCSC genome browser (GRCh37/hg19 version). Genes with a length smaller than 1 kb and overlapping with or less than 2 kb away from other genes were excluded for calculation to avoid enrichment double counting. The empirical cumulative distribution function (ECDF) plot was generated using log2 transformed TRs (averaged from replicates or triplicates) and plotted using R. Wilcoxon rank sum test was used to evaluate the difference between groups.

**Reporting summary**. Further information on research design is available in the Nature Research Reporting Summary linked to this article.

## Data availability
The datasets generated from this study including RNA-Seq, ChIP-Seq, and ATAC-Seq data has been deposited in GEO (GSE153129). Source data are provided with this paper.

## Code availability
The script for calculating travel ratio can be found at github (https://github.com/maxabruzzi/travelRatio) and DOI of https://doi.org/10.5281/zenodo.4383281.

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

## Acknowledgements

This work was supported by grants from the National Institutes of Health (NIH R01AR066530 and R01CA225463) to G.L.S. This publication includes data generated at the UC San Diego IGM Genomics Center utilizing an Illumina NovaSeq 6000 that was purchased with funding from a National Institutes of Health SIG grant (#S10 OD026929). This publication includes data generated at the UC San Diego Center for Epigenomics. Work at the Center for Epigenomics was supported in part by the UC San Diego School of Medicine. This work was also supported by grants from NIH/NCATS (UG3TR003355 and UG3TR002968) to P.G. and S.D.

## Author contributions

J.L. and G.L.S. conceived of the project, designed the experiments, and wrote the paper. J.L., M.T., Y.C., and V.B. performed the experiments. J.L. and X.X. performed the bioinformatics analysis. P.G., S.D., M.F., and J.L. performed the intestinal organoid experiments. J.L., G.L.S., X.X., M.F., V.B., P.T., S.D., and P.G. contributed to the editing and reading of the manuscript.

## Competing interests

The authors declare no competing interests.
