## [Peer Review File · Nature Communications]

REVIEWER COMMENTS

Reviewer #1 (Remarks to the Author):

Li and colleagues study the role of transcriptional elongation in tissue control. Utilizing human epidermal keratinocytes as a model system, the authors show that a large portion of epidermal differentiation genes are subjected to promoter proximal pausing of RNA Pol II during epidermal differentiation. Through loss-of-functions studies the authors identified that transcriptional elongation factor, SPT6, is required for promotion of the epidermal differentiation program. The loss of SPT6 activity resulted in a failure to activate epidermal differentiation genes and led to the induction of non-epidermal factors. This, while many other factors involved in transcriptional elongation were found to be dispensable for epidermal differentiation. Using ChIP-seq analysis, authors next show that SPT6 directly regulates epidermal differentiation genes by promoting their transcriptional elongation. Among SPT6 targets, authors propose the epidermal master regulator p63 as a downstream effector of SPT6. Upon silencing of SPT6, the expression of p63 is dramatically reduced, whereas the expression of non-epidermal factors is induced. Probing into molecular mechanisms the authors show that p63 function to directly repress the expression of non-epidermal factors, and its enforced expression in SPT6-silenced keratinocytes is sufficient to maintain non-epidermal genes at a repressed state. Altogether, this study suggests that SPT6 maintains epidermal differentiation program by promoting transcriptional elongation of epidermal differentiation genes, while blocking the expression on non-epidermal genes in a p63-dependent manner.

The manuscript is well organized, and experiments are very well performed with proper controls and data analysis. Overall, this study will be of great interest to the broad audience of Nature Communications, including readers who are interested in somatic tissue differentiation and development, epigenetic regulation of stem cell function, and transcriptional control. Below are the specific comments to improve the manuscript:

1. The authors show that ~37% (850/2,307) differentiation genes already have RNA Pol II binding in proliferation condition. To exclude the possibility of premature differentiation of analysed cells the authors address this by providing staining for keratin 1 in proliferating conditions. However, this analysis does not adequately address the possibility that these differentiation genes are already expressed in proliferating conditions in lower levels when compared differentiating conditions, hence having RNA Pol II.

Including such analysis comparing between the two groups RNA levels will significantly strengthen that these ~37% of epidermal differentiation genes are bound by paused RNA Pol II.

2. The authors describe that about 1/3 of differentiation genes are poised for activation in proliferation conditions and are characterized by promoter proximal pausing of RNA Pol II, while the remaining epidermal differentiation genes do not have RNA Pol II binding. For the first group of differentiation genes, authors provide gene examples of several key differentiation transcription factors. For the second group of differentiation genes examples do not seem to include transcription factors. Following this notion, it would be interesting to perform some type of GO term analysis for the two groups and check if there is something unique about the first group of "poised" genes. Is there is something special about them such as enrichment for transcription factors? Early vs. late epidermal differentiation stages?

3. Figures 1i-j and S1i-l will benefit from an addition of RNA-seq tracks in proliferating and differentiating conditions (assuming that the current scales used in browser tracks will enable informative view of RNA-seq data).

4. What does the values in all tables showing RNA-seq data represent? FPKM? TPM? Something else?

This is not clear, especially since authors described in methods section that they considered only genes with more than 10 reads – does this take into account each sample's total reads or the transcript length. Some additional clarification is needed here since some genes have close to "0" values in all groups.

5. Using ATAC-seq the authors have determined that SPT6 is closing the genomic regions specifying non-epidermal lineages (Figure 4 data). It is not clear from the presented whether SPT6 directly regulates those non-epidermal genes or if it's a secondary effect. This must be presented more clearly in the text.

6. The authors show that upon knockdown of SPT6 there is an upregulation of non-epidermal genes enriched for several lineages including many genes of the intestinal lineage. While these alterations are very clear and striking, perhaps authors should be more cautious with their statement that "loss of SPT6 also caused the spontaneous transdifferentiation of epidermal cells into an intestinal phenotype". What is the evidence that supports intestinal phenotype? Can this be really concluded based on the analysis of upregulated genes expression alone?

7. Pearson correlation map in Figure 5d should also include tissues analysed in Figure 5a such as Trachea and Adipocytes.

8. Regarding point 6, authors should include some H&E images of SPT6i tissue compared to control. Do they see alterations in cell shape or other characteristics of epidermal cells that could support intestinal phenotype such as microvilli structures?

9. Authors show very nicely that retroviral expression of p63 in SPT6i cells suppressed the expression of key non-epidermal genes. This analysis is missing the effect of p63 expression in SPT6i cells on epidermal differentiation genes. Can the expression of p63 alone rescue the impaired differentiation observed in SPT6i skin model?

10. References for methods should be a part of the manuscript's main "references" section.

11. There is no caption of Figure 2l.

Reviewer #2 (Remarks to the Author):

In this manuscript, authors identified SPT6 as a key factor controlling Pol II elongation during keratinocyte differentiation, and proposed that SPT6 controls the epidermal cell fate by regulating gene transcription of the epidermal master regulator p63. Authors made interesting observation that, for many genes that are induced during differentiation, Pol II is stalled at the promoter and released to elongate and promote transcription of these genes when differentiation is initiated, and that SPT6 is the factor that controls this process. It would have been interesting if authors continue to dissect the mechanism, as pol II pausing/elongation during keratinocyte differentiation is an unexplored topic. However, the link to p63 is strange, as the SPT6-p63 axis does not explain the mechanism of Pol II pausing during epidermal differentiation.

Major points:

1. As SPT6 is important for pol II elongation, one would expect that knock-down of SPT6 will give rise to pol II stalling at the promoter. What is the rationale to perform ATAC-seq analysis upon SPT6 knockdown? Do authors expect that SPT6 cooperate with transcription factors, either activate or

repress gene expression at enhancer? If so, authors did not further explore this question. In authors' analyses, they observed that, upon SPT6 knockdown, both up or down regulated genes have increased accessibility at their promoters. This is fully expected for a factor important for pol II elongation, as there will be more pol II occupancy at the promoters. As described by authors themselves, 'This further supports the idea that the genes dependent on SPT6 for expression is mediated through transcription elongation rather than a closure of chromatin', using ATAC-seq approach cannot understand the mechanism of SPT6-mediated pol II pausing.

2. Following their ATAC-seq analyses, continuing with p63 is kind of cherry picking, as p63 is not the only enriched motifs. In addition, the statement 'These results suggest that SPT6 may be important for promoting the accessibility of P63 binding sites while closing genomic regions specifying non-epidermal lineages' is misleading, as it suggests that SPT6 is a co-regulator of p63, and cooperates with p63 to either open or close chromatin. Actually authors analyses show that SPT6 controls p63 transcription by regulating pol II elongation at the p63 gene body. Once p63 expression is down, the open chromatin regions that normally occupied by p63 is affected. Importantly, p63 is not the only transcription factor affected by SPT6 knockdown. The complete epidermal TF network controlling keratinocyte proliferation and differentiation could be affected. A careful analysis should be performed here.

3. Using 'an intestinal phenotype' might be a bit simplistic. It has been well established that p63 is a master regulator that defines stratified epithelial cells that are different from simple epithelial cells where p63 is not expressed. Overexpression of p63 in simple epithelial cells will give rise to cells that are able to stratify. Intestinal cells are a type of simple epithelial cells. Therefore, it is important for authors to analyse other simple epithelial and stratified epithelial cells with SPT6 knockdown. In addition, author should cite proper literature (e.g. Shalom-Feuerstein et al., Cell Death and Differentiation, 2011; Qu et al., Cell Reports 2018), as the concept that losing p63 results in change of the epithelial cell fate is not new.

4. To analyse pol II pausing during differentiation, authors can compare their data with a previously published pol II ChIP-seq datasets during differentiation (Kouwenhoven et al., EMBOR 2015). This is not only to confirm their findings but also to study dynamics of pol II pausing, as the previous dataset was obtained from a time course.

Detailed points:

1. Pol II travel rate (TR) is represented in 4 blocks. This is not the best way to visualize. It would be better to sort all genes with their TR and make a line plot or scatter plot according to their TR. In this way, TRs of both proliferating and differentiating cells can be plotted together and compared.
2. Bioinformatic analyses in this manuscript should have more rigor. For example, motif analysis should not only have percentage of certain motifs but also show their enrichment with random genomic background, e.g. using HOMER or GIMME motif.
3. Material and methods should be more precise, e.g. which antibodies of pol II are used in this study is important to know.

Reviewer #3 (Remarks to the Author):

This manuscript investigates transcriptional regulation of epidermal differentiation by genome-wide analyses of RNA polymerase II (Pol II) distribution in proliferating versus differentiating cells. The results indicate an important contribution of post-initiation control mechanisms and uncover a specific role for the conserved histone chaperone (and transcription elongation factor) SPT6 in driving

expression of genes needed for differentiation. Loss of SPT6 not only prevents normal skin differentiation and stratification but induces trans-differentiation of epidermal cells to an intestinal phenotype—an effect the authors ascribe to “stalled transcription” of the gene encoding the cell fate-determining p63. These are important findings, and the data generally support the major conclusions. Although it is not terribly novel or surprising that release from a promoter-proximal pause is the rate-limiting step for expression of genes important to a cell fate decision, a specific reliance on SPT6, heretofore thought of as a general regulator, is unexpected. The switching of epidermal cells to the intestinal phenotype in the absence of SPT6 function is also dramatic. Before I can whole-heartedly recommend publication, however, I would like to see the specific requirement for SPT6 better defined, and the mechanistic connection between the transcription elongation function of SPT6 and expression of p63 firmed up. Some revisions of the text would also be advisable, to portray more accurately the current state of knowledge about elongation control and functions of SPT6 therein. Below I list my specific concerns:

1. Although it is important and interesting that many (~30%) of the genes upregulated when epidermal progenitor cells are induced to differentiate are controlled at the elongation stage rather than by Pol II recruitment, in 2020 it is not really surprising; similar findings have been reported for genes involved in differentiation more broadly, and in specific pathways such as cell division control and inflammatory responses. I would advise the authors to tone down the language that suggests a major paradigm shift, e.g. in the abstract (“It is assumed that the rate-limiting step...is the recruitment of Pol II to promoters.”) and elsewhere in the manuscript.
2. In the Introduction, third paragraph, the authors present a somewhat outdated picture of the mechanisms underlying pause release, specifically omitting citations of recent work from the Cramer lab (Vos et al., *Nature* 560: 601-6, 2018), showing that in converting a paused elongation complex to an actively elongating one, P-TEFb phosphorylates more components than the ones listed here, including the Pol II linker region and, most relevant to the present study, SPT6 and subunits of the PAF complex.
3. Figs. 1d and 1e are redundant, displaying the same results in different formats; simply adding the percentages shown in brackets in 1e to the diagram in 1d would obviate the need for two panels (a minor point).
4. The vertical arrangement of the browser tracks in Fig. 1i-j and Supp. Fig. 1i-l is needlessly confusing, juxtaposing the Ser2 tracks from proliferating and differentiating cells to allow easy comparison but not the Ser5 tracks. I’m not sure why the authors chose this order but I would suggest grouping the tracks by antibody (minor point).
5. In general, the analyses of Ser2 and Ser5 phosphorylation do not add much to the story. To my eye, it appears that Ser2 is increasing along most activated genes in proportion to total Pol II, at least until transcription reaches the termination zone. Likewise, the effects on Ser5 are mostly correlated with the total Pol II occupancy, i.e., dropping over the promoter-proximal region when Pol II is released into productive elongation.
6. In discussing the data in Fig. 1, the authors note that the ~37% of pause-regulated induced genes include transcription factors that promote differentiation. An obvious but important question that the authors do not address is whether the targets of those factors are enriched within the ~63% of genes that depend on Pol II recruitment for their induction.
7. The “small RNAi screen” done to evaluate the involvement of elongation regulators in differentiation did not include subunits of the PAF complex (see point 2 above) or NELF. Efficiency of knockdown, moreover, was assessed by measuring levels of mRNA rather than protein (Fig. 2a). Because the specific dependence on SPT6 would be a major take-home message, it really needs to be established with more certainty.
8. The ~30% overlap between SPT6-dependent and differentiation genes is significant (Fig. 2i,k, Supp. Table 6), but have the authors further broken down their gene sets into paused and non-paused genes to see if the correlation becomes even stronger?
9. The data in Supplemental Fig. 2c-e are quite central to the story and supportive of the authors’

model, and probably belong in a main figure.

10. In Results, under sub-heading "SPT6 suppresses an intestinal fate...", the authors should probably avoid the use of the word "incredibly" to describe their results.

11. The functional connection between SPT6 and p63 remains unclear to me. On the one hand, depletion of SPT6 results in decreased chromatin accessibility (measured by ATAC-seq) at promoters with p63-binding sites (perhaps implicating the histone chaperone function of SPT6 in this effect). On the other, SPT6 depletion leads to a (substantial) reduction in p63 mRNA (Fig. 2g) and a (much more subtle) loss of Pol II occupancy on the apparently pause-regulated P63 gene (Fig. 5e). I am not entirely convinced by this single snapshot that SPT6 is required for efficient elongation at this locus. Moreover, the loss of chromatin accessibility at P63-regulated promoters suggests an alternative mechanism by which SPT6 depletion might mimic p63 loss. The rescue of "SPT6i" effects on gene expression by ectopic p63 expression (Fig. 5g-h) is impressive but does not discriminate between those mechanisms. The authors should at least acknowledge and discuss these uncertainties but should perhaps consider additional experiments to strengthen the causal connection between SPT6's elongation-promoting function and p63 expression (e.g. some measurement of nascent transcription).

12. The Discussion is simply a recap in abbreviated, data-free form of the Results section. I would prefer that the authors take the opportunity to place their work in context of other studies of transcription elongation control and epidermal differentiation, discuss or even speculate about how SPT6 might be specifically required for this regulation, and address some of the unanswered questions including but not necessarily limited to the ones I raised in point 11.

We thank the reviewers for their suggestions that have made this manuscript substantially stronger. We have now addressed all of the reviewers' concerns via additional experiments, further analysis, or better clarification. Please see below for a point-by-point response to each reviewer's question. New text written in the manuscript as a response to reviewer questions are underlined.

Reviewer #1

We thank the Reviewer for helpful and positive comments on the paper, which we believe have provided valuable guidance in improving the paper.

Reviewer #1 (Remarks to the Author):

Li and colleagues study the role of transcriptional elongation in tissue control. Utilizing human epidermal keratinocytes as a model system, the authors show that a large portion of epidermal differentiation genes are subjected to promoter proximal pausing of RNA Pol II during epidermal differentiation. Through loss-of-functions studies the authors identified that transcriptional elongation factor, SPT6, is required for promotion of the epidermal differentiation program. The loss of SPT6 activity resulted in a failure to activate epidermal differentiation genes and led to the induction of non-epidermal factors. This, while many other factors involved in transcriptional elongation were found to be dispensable for epidermal differentiation. Using ChIP-seq analysis, authors next show that SPT6 directly regulates epidermal differentiation genes by promoting their transcriptional elongation. Among SPT6 targets, authors propose the epidermal master regulator p63 as a downstream effector of SPT6. Upon silencing of SPT6, the expression of p63 is dramatically reduced, whereas the expression of non-epidermal factors is induced. Probing into molecular mechanisms the authors show that p63 function to directly repress the expression of non-epidermal factors, and its enforced expression in SPT6-silenced keratinocytes is sufficient to maintain non-epidermal genes at a repressed state. Altogether, this study suggests that SPT6 maintains epidermal differentiation program by promoting transcriptional elongation of epidermal differentiation genes, while blocking the expression on non-epidermal genes in a p63-dependent manner. The manuscript is well organized, and experiments are very well performed with proper controls and data analysis. Overall, this study will be of great interest to the broad audience of Nature Communications, including readers who are interested in somatic tissue differentiation and development, epigenetic regulation of stem cell function, and transcriptional control. Below are the specific comments to improve the manuscript:

Reviewer #1 remarks to the authors:

1. The authors show that ~37% (850/2,307) differentiation genes already have RNA Pol II binding in proliferation condition. To exclude the possibility of premature differentiation of analysed cells the authors address this by providing staining for keratin 1 in proliferating conditions. However, this analysis does not adequately address the possibility that these differentiation genes are already expressed in proliferating conditions in lower levels when compared differentiating conditions, hence having RNA Pol II. Including such analysis comparing between the two groups RNA levels will significantly strengthen that these ~37% of epidermal differentiation genes are bound by paused RNA Pol II.

We thank the reviewer for this question. The current literature suggests that >99% of paused Pol II regulated genes are already expressed to some extent as measured by

nascent RNA measurement assays. This is summarized in a recent review by Leighton Core and Karen Adelman which they stated as, “However, paused Pol II is very infrequently found at genes that are not expressed (<1% of paused genes are deemed inactive by nascent RNA-seq techniques), suggesting that pause release is an inherently leaky process (Core and Adelman, 2019).” Interestingly as the reviewer suggested, this question hasn’t been addressed in epidermal keratinocytes so we divided the RPKM gene expression levels of the induced differentiation genes (with Pol II binding in proliferation conditions) into the following 3 categories: 1) Percent of genes with RPKM values between 0-50 (low to no expression). 2) Percent of genes with RPKM values between 50-100 (medium expression). 3) Percent of genes with RPKM values >100 (higher expression). As shown in Supplementary Figure 1h, 99.35% of the highly paused, 95.92% of the moderately paused, and 95.51% of non-paused differentiation genes had RPKM values from 0-50. This suggests that the vast majority of these genes have low to no expression. Furthermore, the distribution of the gene expression is also correlated with the amount of pausing. For example, in the 155 highly paused genes there were no genes that had RPKM values over 100 and only 0.65% were between 50-100. In the moderately paused genes 1.86% of the genes were between 50-100 RPKM and 2.23% were greater than 100 RPKM. The 156 non-paused genes had the greatest percentage (3.85%) of genes expressing RPKM levels higher than 100.

2. The authors describe that about 1/3 of differentiation genes are poised for activation in proliferation conditions and are characterized by promoter proximal pausing of RNA Pol II, while the remaining epidermal differentiation genes do not have RNA Pol II binding. For the first group of differentiation genes, authors provide gene examples of several key differentiation transcription factors. For the second group of differentiation genes examples do not seem to include transcription factors. Following this notion, it would be interesting to perform some type of GO term analysis for the two groups and check if there is something unique about the first group of “poised” genes. Is there is something special about them such as enrichment for transcription factors? Early vs. late epidermal differentiation stages?

This is an interesting question and we approached this in 2 ways. First, we performed gene ontology on the 694 paused and 1,457 (without Pol II loaded) differentiation genes. The paused differentiation genes were enriched for “regulators of transcription and gene expression” while the other 1,457 differentiation genes did not enrich for those terms (Supplementary Fig. 2a-b). Second, we manually looked for transcription factors previously published to promote epidermal differentiation in both gene sets. Interestingly, there were 14 transcription factors found in the highly and moderately paused genes (694 genes) including GRHL1, CEBPG, MAFB, POU2F3, RORA, POU3F1, KLF4, MAF, FOSL2, DLX3, DLX5, CEBPB, KLF5, and OVOL1 (Supplementary Figure 2c). There were only 4 transcription factors found in the induced differentiation genes (1,457 genes) that did not have Pol II binding in proliferative cells (Supplementary Figure 2d). Thus, the paused genes had a 3.5 fold enrichment (14 versus 4) in transcription factors known to be important for epidermal differentiation even though it had 50% less overall genes (694 versus 1457 genes).

3. Figures 1i-j and S1i-l will benefit from an addition of RNA-seq tracks in proliferating and differentiating conditions (assuming that the current scales used in browser tracks will enable informative view of RNA-seq data).

We have now added the RNA-seq data and it shows that there is no mRNA expression of these differentiation genes in progenitor cells. Expression of these genes only occurs upon differentiation. This is now shown in Figure 1h-i and Supplementary Figure 1k-n.

4. What do the values in all tables showing RNA-seq data represent? FPKM? TPM? Something else? This is not clear, especially since authors described in methods section that they considered only genes with more than 10 reads – does this take into account each sample's total reads or the transcript length. Some additional clarification is needed here since some genes have close to "0" values in all groups.

The values are Log2 transformed RPKM values. This has now been labeled in the supplementary tables with RNA-Seq data.

5. Using ATAC-seq the authors have determined that SPT6 is closing the genomic regions specifying non-epidermal lineages (Figure 4 data). It is not clear from the presented whether SPT6 directly regulates those non-epidermal genes or if it's a secondary effect. This must be presented more clearly in the text.

SPT6 does not directly regulate the non-epidermal genes. SPT6 promotes the transcriptional elongation of P63. Without SPT6, P63 expression is attenuated. P63 binds and represses non-epidermal genes such as those important for intestinal fate such as CDHR2, EPCAM, HNF1A, and FOXA1. Thus, without transcription of P63, these intestinal genes are upregulated. We have now written this in the text explaining that SPT6 regulation of non-epidermal genes is a secondary effect due to P63 regulation. This is now stated as, "These results suggest that SPT6 may potentially be regulating the transcription elongation of P63 and the other pause regulated transcription factors such as FOSL2 and KLF5. Thus without SPT6, P63 and the other epidermal transcription factor gene expression may go down which would then lead to closure of their bound sites. The increase in chromatin accessibility at non-epidermal lineages may be a secondary effect of SPT6 regulation. SPT6 may directly regulate factors such as P63 and these transcription factors may in turn regulate non-epidermal genes."

6. The authors show that upon knockdown of SPT6 there is an upregulation of non-epidermal genes enriched for several lineages including many genes of the intestinal lineage. While these alterations are very clear and striking, perhaps authors should be more cautious with their statement that "loss of SPT6 also caused the spontaneous transdifferentiation of epidermal cells into an intestinal phenotype". What is the evidence that supports intestinal phenotype? Can this be really concluded based on the analysis of upregulated genes expression alone?

We agree with the reviewer and performed additional experiments to determine if loss of SPT6 leads to an intestinal phenotype. To address this, we collaborated with Pradipta Ghosh's lab at UCSD who works on generating intestinal organoids from colon as well as IPS cells. We wanted to determine if SPT6 knockdown cells could generate "intestinal organoids" by expanding from single cells in matrigel. ~90% of control cells formed large clumps of cells that were solid spheres which has no resemblance to intestinal organoids (Fig. 7a, d-f). Human colon organoids (these were derived from healthy adult human subjects undergoing routine colonoscopy) are characterized by centered lumen and

budding crypt structures (Fig. 7d). Importantly, SPT6 knockdown cells formed structures that resembled human colon organoids with centered lumen as well as budding crypt-like domains (Fig. 7b-d). Budding was detected up to ~15% of SPT6 knockdown organoids while none was observed in control cells (Fig. 7g). Loss of SPT6 in the organoids also caused the expression of intestinal genes (Fig. 7h). These results suggest that SPT6 depleted cells can result in an intestinal phenotype. We believe that a further more comprehensive analysis of the intestinal phenotype is beyond the scope of this manuscript as we are currently trying to improve the protocol as well as the addition of other transcription factors to fully convert epidermal cells into intestinal cells. A more comprehensive analysis and potential full conversion into intestinal cells with additional factors will be presented in a future manuscript.

7. Pearson correlation map in Figure 5d should also include tissues analysed in Figure 5a such as Trachea and Adipocytes.

We have now included this additional analysis in the new Figure 6d and it shows that SPT6 depletion is still the most significantly correlated with the colon signature. There is no real correlation with adipocyte (0.125), and trachea (0.07).

8. Regarding point 6, authors should include some H&E images of SPT6i tissue compared to control. Do they see alterations in cell shape or other characteristics of epidermal cells that could support intestinal phenotype such as microvilli structures?

We have now performed the experiment and SPT6 depletion leads to a poorly stratified tissue and a complete absence of stratum corneum formation (Figure 2h). We did not see microvilli structures since the conditions used are not optimal for intestinal differentiation. However as described in the answer to question #6, we were able to culture SPT6 knockdown cells that resembled intestinal organoids as shown in Figure 7.

9. Authors show very nicely that retroviral expression of p63 in SPT6i cells suppressed the expression of key non-epidermal genes. This analysis is missing the effect of p63 expression in SPT6i cells on epidermal differentiation genes. Can the expression of p63 alone rescue the impaired differentiation observed in SPT6i skin model?

This is an important question and we have performed this experiment. Expression of P63 did not rescue the impaired differentiation observed in SPT6 depleted skin (Figure 6j). This is due to SPT6 directly regulating these differentiation genes through transcription elongation (Figure 3).

10. References for methods should be a part of the manuscript's main "references" section.

This is now included in the main manuscript.

11. There is no caption of Figure 2l.

This has now been fixed and now represented as Figure 2n.

Reviewer #2

We thank the Reviewer for thoughtful and constructive comments on the paper as well as the experiments the reviewer suggested. We have made efforts to address these as follows.

Reviewer #2 (Remarks to the Author):

In this manuscript, authors identified SPT6 as a key factor controlling Pol II elongation during keratinocyte differentiation, and proposed that SPT6 controls the epidermal cell fate by regulating gene transcription of the epidermal master regulator p63. Authors made interesting observation that, for many genes that are induced during differentiation, Pol II is stalled at the promoter and released to elongate and promote transcription of these genes when differentiation is initiated, and that SPT6 is the factor that is controls this process. It would have been interesting if authors continue to dissect the mechanism, as pol II pausing/elongation during keratinocyte differentiation is an unexplored topic. However, the link to p63 is strange, as the SPT6-p63 axis does not explain the mechanism of Pol II pausing during epidermal differentiation.

We have now further explored the mechanism of Pol II pausing during epidermal differentiation with the addition of PAF1 data that is now described in Major Point #1 and shown in Figure 4.

Major points:

1. As SPT6 is important for pol II elongation, one would expect that knock-down of SPT6 will give rise to pol II stalling at the promoter. What is the rationale to perform ATAC-seq analysis upon SPT6 knockdown? Do authors expect that SPT6 cooperate with transcription factors, either activate or repress gene expression at enhancer? If so, authors did not further explore this question. In authors' analyses, they observed that, upon SPT6 knockdown, both up or down regulated genes have increased accessibility at their promoters. This is fully expected for a factor important for pol II elongation, as there will be more pol II occupancy at the promoters. As described by authors themselves, 'This further supports the idea that the genes dependent on SPT6 for expression is mediated through transcription elongation rather than a closure of chromatin', using ATAC-seq approach cannot understand the mechanism of SPT6-mediated pol II pausing.

We thank the reviewer for this important question. We performed ATAC-Seq upon SPT6 knockdown for 2 reasons. First, we wanted to determine if loss of transcription elongation on differentiation genes due to SPT6 depletion led to altered chromatin accessibility at those genes. This is not an unreasonable question to ask since there is a possibility that the presence of stalled Pol II could lead to secondary effects such as alterations in chromatin accessibility. Furthermore this topic has not been explored in primary human cells. Second, there was a previous publication that SPT6 could promote H3K27 demethylation and thus knockdown of SPT6 would result in increased repressive chromatin (H3K27me3)(Wang et al., 2013). Thus, it was possible that loss of SPT6 would have resulted in decreased chromatin accessibility at epidermal differentiation genes. However we didn't observe this as there was increased chromatin accessibility. We have now stated this more clearly in the text.

We agree with the reviewer that ATAC-Seq doesn't unravel how SPT6 mediates Pol II elongation. We have now made progress in unraveling its mechanism. The association of PAF and SPT6 with Pol II (shown by cryo-EM) has provided the molecular basis of Pol II pause release and elongation in-vitro(Vos et al., 2018). It is currently unclear whether this same mode of regulation occurs in a tissue setting. Notably, knockdown of

PAF1 phenocopied SPT6 depletion. PAF1 loss blocked epidermal differentiation in regenerated human skin (Figure 4a-c). PAF1 bound to the same differentiation genes as SPT6 (Fig. 4d). Loss of PAF1 also resulted in accumulation of Pol II at the transcription start site of the differentiation genes suggesting its importance for the elongation of these genes (Figure 4e). Our data suggests that SPT6 promotes elongation in conjunction with PAF1 to promote differentiation in a tissue setting which is in line with the in-vitro model of elongation.

2. Following their ATAC-seq analyses, continuing with p63 is kind of cherry picking, as p63 is not the only enriched motifs. In addition, the statement 'These results suggest that SPT6 may be important for promoting the accessibility of P63 binding sites while closing genomic regions specifying non-epidermal lineages' is misleading, as it suggests that SPT6 is a co-regulator of p63, and cooperates with p63 to either open or close chromatin. Actually authors analyses show that SPT6 controls p63 transcription by regulating pol II elongation at the p63 gene body. Once p63 expression is down, the open chromatin regions that normally occupied by p63 is affected. Importantly, p63 is not the only transcription factor affected by SPT6 knockdown. The complete epidermal TF network controlling keratinocyte proliferation and differentiation could be affected. A careful analysis should be performed here.

We don't believe that P63 was cherry picked as it was the most significantly enriched transcription factor binding site found in the less accessible regions (Figure 5e). We agree with the reviewer that P63 is not the only transcription factor affected by SPT6 knockdown. These other pause regulated transcription factor includes KLF5 and FOSL2. We have now stated this in the text. However, the further characterization of these additional factors is beyond the scope of this manuscript since ChIP-Seq/RNA-Seq data in primary human keratinocytes is not available for these other factors. In contrast P63 is a well studied/characterized protein with publicly available data in primary human keratinocytes.

We have also rephrased how SPT6 is regulating P63 to make it more clear and is now stated as, "These results suggest that SPT6 may potentially be regulating the transcription elongation of P63 and the other pause regulated transcription factors such as FOSL2 and KLF5. Thus without SPT6, P63 and the other epidermal transcription factor gene expression may go down which would then lead to closure of their bound sites. The increase in chromatin accessibility at non-epidermal lineages may be a secondary effect of SPT6 regulation. SPT6 may directly regulate factors such as P63 and these transcription factors may in turn regulate non-epidermal genes."

3. Using 'an intestinal phenotype' might be a bit simplistic. It has been well established that p63 is a master regulator that defines stratified epithelial cells that are different from simple epithelial cells where p63 is not expressed. Overexpression of p63 in simple epithelial cells will give rise to cells that are able to stratify. Intestinal cells are a type of simple epithelial cells. Therefore, it is important for authors to analyse other simple epithelial and stratified epithelial cells with SPT6 knockdown. In addition, author should cite proper literature (e.g. Shalom-Feuerstein et al., Cell Death and Differentiation, 2011; Qu et al., Cell Reports 2018), as the concept that losing p63 results in change of the epithelial cell fate is not new.

We thank the reviewer for the suggestion to analyze other simple epithelial and stratified epithelial cells. We believe this specific experiment is beyond the scope of the manuscript as our paper is focused on primary human epidermal keratinocytes and not on other types of epithelia. Furthermore, it is not trivial to obtain other types of primary human simple and stratified epithelial cells. It is also not trivial to obtain the endogenous substrate that these cells depend on to be able to regenerate their respective tissue. Epidermis is also one of the few tissues you can regenerate in cell culture while the protocols to regenerate other simple or stratified tissue has not been worked out. We would also have to work out how to knockdown genes in these other types of primary epithelial cells.

However, we agree with the reviewer that additional experiments need to be performed to determine if loss of SPT6 leads to a more “intestinal phenotype”. To address this, we collaborated with Pradipta Ghosh’s lab at UCSD who works on generating intestinal organoids from colon as well as IPS cells. We wanted to determine if SPT6 knockdown cells could generate “intestinal organoids” by expanding from single cells in matrigel. ~90% of control cells formed large clumps of cells that were solid spheres which has no resemblance to intestinal organoids (Fig. 7a, d-f). Human colon organoids (these were derived from healthy adult human subjects undergoing routine colonoscopy) are characterized by centered lumen and budding crypt structures (Fig. 7d). Importantly, SPT6 knockdown cells formed structures that resembled human colon organoids with centered lumen as well as budding crypt-like domains (Fig. 7b-d). Budding was detected up to ~15% of SPT6 knockdown organoids while none was observed in control cells (Fig. 7g). Loss of SPT6 in the organoids also caused the expression of intestinal genes (Fig. 7h). These results suggest that SPT6 depleted cells can result in an intestinal phenotype. We believe that a further more comprehensive analysis of the intestinal phenotype is beyond the scope of this manuscript as we are currently trying to improve the protocol as well as the addition of other transcription factors to fully convert epidermal cells into intestinal cells. A more comprehensive analysis and potential full conversion into intestinal cells with additional factors will be presented in a future manuscript.

As the reviewer suggested, we have now cited the proper literature on P63 in the discussion section. It is now stated as, “These results are consistent with observations that p63 null mouse embryos develop intestine-like metaplasia⁵³. Another group also showed that in embryonic epithelia, p63 knockout prevented stratification which resulted in a single layer epithelium with expression of mesodermal/muscle genes⁶⁹. More recently, Qu et al. used keratinocytes derived from EEC patients to show that epidermal cell identity was lost in cells with p63 DNA binding mutations⁷⁰. These mutant cells also had upregulated expression of neuronal and mesenchymal genes. It is notable that SPT6 loss had the highest Pearson correlation with colon and followed by skeletal muscle suggesting regulation of P63 may be partially responsible for some of its phenotype. In support of this, over 45% of the genes upregulated upon SPT6 knockdown had P63 binding suggesting that P63 may be acting as a repressor on these genes.”

4. To analyse pol II pausing during differentiation, authors can compare their data with a previously published pol II ChIP-seq datasets during differentiation (Kouwenhoven et al., EMBOR 2015). This is not only to confirm their findings but also to study dynamics of pol

II pausing, as the previous dataset was obtained from a time course.

We thank the reviewer for this suggestion and the new comparison has now been added to Supplementary Figure 1b, 1i-j and show similar results to our study.

Detailed points:

1. Pol II travel rate (TR) is represented in 4 blocks. This is not the best way to visualize. It would be better to sort all genes with their TR and make a line plot or scatter plot according to their TR. In this way, TRs of both proliferating and differentiating cells can be plotted together and compared.

We thank the reviewer for this suggestion but we wanted to plot the travel ratio in the same way as others in the field have already published and accepted as the standard (Chen et al., 2015; Day et al., 2016; Rahl et al., 2010; Yu et al., 2015). Our plots also show both proliferation and differentiation samples all plotted together and compared as the reviewer suggested (Figure 1g). Our analysis in Figure 1c also allows us to categorize the genes as either “highly, moderately, or non-paused” which then allows us to determine how the highly and moderately paused genes change in travel ratio upon differentiation (Figure 1c, 1g).

2. Bioinformatic analyses in this manuscript should have more rigor. For example, motif analysis should not only have percentage of certain motifs but also show their enrichment with random genomic background, e.g. using HOMER or GIMME motif.

This information has now been added and shown in Figure 5e,5g.

3. Material and methods should be more precise, e.g. which antibodies of pol II are used in this study is important to know.

This information along with all the antibody catalog numbers are now included in the new methods section.

Reviewer #3

We thank the Reviewer for the insightful comments which we have used to substantially strengthen the manuscript.

Reviewer #3 (Remarks to the Author):

This manuscript investigates transcriptional regulation of epidermal differentiation by genome-wide analyses of RNA polymerase II (Pol II) distribution in proliferating versus differentiating cells. The results indicate an important contribution of post-initiation control mechanisms and uncover a specific role for the conserved histone chaperone (and transcription elongation factor) SPT6 in driving expression of genes needed for differentiation. Loss of SPT6 not only prevents normal skin differentiation and stratification but induces trans-differentiation of epidermal cells to an intestinal phenotype—an effect the authors ascribe to “stalled transcription” of the gene encoding the cell fate-determining p63. These are important findings, and the data generally support the major conclusions. Although it is not terribly novel or surprising that release from a promoter-proximal pause is the rate-limiting step for expression of genes important to a cell fate decision, a specific reliance on SPT6, heretofore thought of as a general regulator, is unexpected. The switching of epidermal cells to the intestinal

phenotype in the absence of SPT6 function is also dramatic. Before I can wholeheartedly recommend publication, however, I would like to see the specific requirement for SPT6 better defined, and the mechanistic connection between the transcription elongation function of SPT6 and expression of p63 firmed up. Some revisions of the text would also be advisable, to portray more accurately the current state of knowledge about elongation control and functions of SPT6 therein. Below I list my specific concerns:

Reviewer #3 remarks to the authors:

1. Although it is important and interesting that many (~30%) of the genes upregulated when epidermal progenitor cells are induced to differentiate are controlled at the elongation stage rather than by Pol II recruitment, in 2020 it is not really surprising; similar findings have been reported for genes involved in differentiation more broadly, and in specific pathways such as cell division control and inflammatory responses. I would advise the authors to tone down the language that suggests a major paradigm shift, e.g. in the abstract (“It is assumed that the rate-limiting step...is the recruitment of Pol II to promoters.”) and elsewhere in the manuscript.

We appreciate this suggestion from the reviewer and have now toned down the language in the manuscript.

2. In the Introduction, third paragraph, the authors present a somewhat outdated picture of the mechanisms underlying pause release, specifically omitting citations of recent work from the Cramer lab (Vos et al., Nature 560: 601-6, 2018), showing that in converting a paused elongation complex to an actively elongating one, P-TEFb phosphorylates more components than the ones listed here, including the Pol II linker region and, most relevant to the present study, SPT6 and subunits of the PAF complex.

We thank the reviewer for this and have now updated our introduction with the newest work as pointed out by the reviewer.

3. Figs. 1d and 1e are redundant, displaying the same results in different formats; simply adding the percentages shown in brackets in 1e to the diagram in 1d would obviate the need for two panels (a minor point).

We have now combined the two panels into one (Figure 1d).

4. The vertical arrangement of the browser tracks in Fig. 1i-j and Supp. Fig. 1i-l is needlessly confusing, juxtaposing the Ser2 tracks from proliferating and differentiating cells to allow easy comparison but not the Ser5 tracks. I'm not sure why the authors chose this order but I would suggest grouping the tracks by antibody (minor point).

We decided to remove the Ser2/Ser5 data since it was redundant with the total Pol II data.

5. In general, the analyses of Ser2 and Ser5 phosphorylation do not add much to the story. To my eye, it appears that Ser2 is increasing along most activated genes in proportion to total Pol II, at least until transcription reaches the termination zone. Likewise, the effects on Ser5 are mostly correlated with the total Pol II occupancy, i.e., dropping over the promoter-proximal region when Pol II is released into productive

elongation.

We agree with the reviewer and decided to remove the Ser2/Ser5 data since it was redundant with the total Pol II data.

6. In discussing the data in Fig. 1, the authors note that the ~37% of pause-regulated induced genes include transcription factors that promote differentiation. An obvious but important question that the authors do not address is whether the targets of those factors are enriched within the ~63% of genes that depend on Pol II recruitment for their induction.

To definitively show that the paused genes was indeed enriched for transcription factors that promoted differentiation, we manually looked for known transcription factors that are necessary for epidermal differentiation in both the genes lists. Importantly, we found that in the moderately and highly paused genes there were 14 known transcription factors necessary for epidermal differentiation including GRHL1, CEBPG, MAFB, POU2F3, RORA, POU3F1, KLF4, MAF, FOSL2, DLX3, DLX5, CEBPB, KLF5, and OVOL1. In contrast, the 1457 induced differentiation genes without Pol II bound in proliferation conditions only had 4 transcription factors. Thus, the paused genes had a 3.5 fold enrichment (14 versus 4) in transcription factors known to be important for epidermal differentiation even though it had 50% less overall genes (694 versus 1457 genes). This is now shown in Supplementary Figure 2c-d.

We also performed gene ontology on the 694 paused and 1,457 (without Pol II loaded) differentiation genes. The paused differentiation genes were enriched for “regulators of transcription and gene expression” while the other 1,457 differentiation genes did not enrich for those terms (Supplementary Figure 2a-b). This corroborates our findings that there was a 3.5 fold increase in transcription factors in the paused differentiation genes versus the differentiation genes not regulated by paused Pol II.

As the reviewer alluded to, this suggests the possibility that the 14 transcription factors in the paused regulated gene set may be turning on the other ~63% of the genes that depend on Pol II recruitment during differentiation. Unfortunately, there isn't publicly available ChIP-Seq data on the 14 epidermal differentiation transcription factors (listed in the above paragraphs) to determine if they directly bind to the 1,457 differentiation genes. However, we were able to use Enrichr to determine the top transcription factors these 1,457 genes were most often co-expressed with (Kuleshov et al., 2016). As shown in Supplementary Figure 2e, these genes were co-expressed with transcription factors such as OVOL1, KLF5, and the GRHL family of transcription factors which were part of the paused genes. This suggests that the 63% of the differentiation genes that depend on Pol II recruitment may be regulated by these pause regulated transcription factors.

7. The “small RNAi screen” done to evaluate the involvement of elongation regulators in differentiation did not include subunits of the PAF complex (see point 2 above) or NELF. Efficiency of knockdown, moreover, was assessed by measuring levels of mRNA rather than protein (Fig. 2a). Because the specific dependence on SPT6 would be a major take-home message, it really needs to be established with more certainty.

Due to the expense and the variability of whether an antibody will work we purchased commercial antibodies targeting ELL, SPT5, and AFF4. The AFF4 antibody we purchased did not give any specific bands but with ELL and SPT6 we observed good

knockdown of each gene on the protein level. This is now shown in Figure 2b where there is significant knockdown of each of these genes on the protein level.

We thank the reviewer for pointing out the PAF complex and the recent cryo-EM study. The association of PAF and SPT6 with Pol II has provided the molecular basis of Pol II pause release and elongation in-vitro. However whether these factors impact tissue differentiation and elongation in-vivo is not known. Because of this we were inspired by the reviewers question to test this. Notably, knockdown PAF1 phenocopied SPT6 depletion. PAF1 loss blocked epidermal differentiation in regenerated human skin (Figure 4a-c). PAF1 bound to the same differentiation genes as SPT6 (Figure 4d). Loss of PAF1 also resulted in accumulation of Pol II at the transcription start site of the differentiation genes suggesting its importance for the elongation of these genes(Figure 4e). Our data suggests that SPT6 promotes elongation in conjunction with PAF1 to promote differentiation in a tissue setting which is in line with the in-vitro model of elongation.

8. The ~30% overlap between SPT6-dependent and differentiation genes is significant (Fig. 2i,k, Supp. Table 6), but have the authors further broken down their gene sets into paused and non-paused genes to see if the correlation becomes even stronger?

We have now performed this analysis and yes the correlation is even stronger. ~44% (303/694) of the paused genes were differentially regulated upon SPT6 knockdown (SPT6i RNA-Seq) (Supplementary Fig. 2f). ~38% (559/1457) of the non-paused genes (induced differentiation genes that depend on Pol II recruitment for their induction) were differentially regulated upon SPT6 knockdown (SPT6i RNA-Seq) (Supplementary Fig. 2g). This suggests that this second group of genes (1457 non-paused) may be regulated by the transcription factors that are pause regulated (Supplementary Fig. 2).

9. The data in Supplemental Fig. 2c-e are quite central to the story and supportive of the authors' model, and probably belong in a main figure.

This has now been placed in the main figure as Figure 3e-g.

10. In Results, under sub-heading "SPT6 suppresses an intestinal fate...", the authors should probably avoid the use of the word "incredibly" to describe their results.

The word "incredibly" has been removed.

11. The functional connection between SPT6 and p63 remains unclear to me. On the one hand, depletion of SPT6 results in decreased chromatin accessibility (measured by ATAC-seq) at promoters with p63-binding sites (perhaps implicating the histone chaperone function of SPT6 in this effect). On the other, SPT6 depletion leads to a (substantial) reduction in p63 mRNA (Fig. 2g) and a (much more subtle) loss of Pol II occupancy on the apparently pause-regulated P63 gene (Fig. 5e). I am not entirely convinced by this single snapshot that SPT6 is required for efficient elongation at this locus. Moreover, the loss of chromatin accessibility at P63-regulated promoters suggests an alternative mechanism by which SPT6 depletion might mimic p63 loss. The rescue of "SPT6i" effects on gene expression by ectopic p63 expression (Fig. 5g-h) is impressive but does not discriminate between those mechanisms. The authors should at least acknowledge and discuss these uncertainties but should perhaps consider additional experiments to strengthen the causal connection between SPT6's elongation-promoting

function and p63 expression (e.g. some measurement of nascent transcription).

We thank the reviewer for this suggestion and have now measured nascent *P63* mRNA levels after SPT6 knockdown. Differentiated control and SPT6 knockdown keratinocytes were pulsed with 5-ethynyl uridine (EU). After the pulse, nascent RNA labeled with EU was used in a copper catalyzed click reaction with azide-modified biotin. The EU-biotin labeled nascent RNA was then purified and selectively isolated using streptavidin magnetic beads. Loss of SPT6 resulted in a significant decrease of nascent *P63* mRNA from the gene body which accumulated at the transcriptional start site. These results suggest that SPT6 is necessary for *P63* mRNA elongation. This is now shown in Figure 6f.

12. The Discussion is simply a recap in abbreviated, data-free form of the Results section. I would prefer that the authors take the opportunity to place their work in context of other studies of transcription elongation control and epidermal differentiation, discuss or even speculate about how SPT6 might be specifically required for this regulation, and address some of the unanswered questions including but not necessarily limited to the ones I raised in point 11.

We have now done this in the new discussion.

REFERENCES

- Chen, F.X., Woodfin, A.R., Gardini, A., Rickels, R.A., Marshall, S.A., Smith, E.R., Shiekhhattar, R., and Shilatifard, A. (2015). PAF1, a Molecular Regulator of Promoter-Proximal Pausing by RNA Polymerase II. *Cell* *162*, 1003-1015.
- Core, L., and Adelman, K. (2019). Promoter-proximal pausing of RNA polymerase II: a nexus of gene regulation. *Genes Dev* *33*, 960-982.
- Day, D.S., Zhang, B., Stevens, S.M., Ferrari, F., Larschan, E.N., Park, P.J., and Pu, W.T. (2016). Comprehensive analysis of promoter-proximal RNA polymerase II pausing across mammalian cell types. *Genome biology* *17*, 120.
- Kuleshov, M.V., Jones, M.R., Rouillard, A.D., Fernandez, N.F., Duan, Q., Wang, Z., Koplev, S., Jenkins, S.L., Jagodnik, K.M., Lachmann, A., *et al.* (2016). Enrichr: a comprehensive gene set enrichment analysis web server 2016 update. *Nucleic Acids Res* *44*, W90-97.
- Rahl, P.B., Lin, C.Y., Seila, A.C., Flynn, R.A., McCuine, S., Burge, C.B., Sharp, P.A., and Young, R.A. (2010). c-Myc regulates transcriptional pause release. *Cell* *141*, 432-445.
- Vos, S.M., Farnung, L., Boehning, M., Wigge, C., Linden, A., Urlaub, H., and Cramer, P. (2018). Structure of activated transcription complex Pol II-DSIF-PAF-SPT6. *Nature* *560*, 607-612.
- Wang, A.H., Zare, H., Mousavi, K., Wang, C., Moravec, C.E., Sirotkin, H.I., Ge, K., Gutierrez-Cruz, G., and Sartorelli, V. (2013). The histone chaperone Spt6 coordinates histone H3K27 demethylation and myogenesis. *EMBO J* *32*, 1075-1086.
- Yu, M., Yang, W., Ni, T., Tang, Z., Nakadai, T., Zhu, J., and Roeder, R.G. (2015). RNA polymerase II-associated factor 1 regulates the release and phosphorylation of paused RNA polymerase II. *Science* *350*, 1383-1386.

REVIEWERS' COMMENTS

Reviewer #1 (Remarks to the Author):

Authors have addressed all comments and concerns. I find the revised manuscript much improved and suitable for publication.

Reviewer #2 (Remarks to the Author):

This referee appreciates authors' great effort to address all questions raised by referees. Most of my concerns have been addressed and the manuscript has significantly improved. Several improvements relevant to my previous comments:

1. The manuscript has a better focus, shifted more on the mechanism of pol2 pausing and elongation. Especially including the new work on PAF1 has strengthen the manuscript.
2. The rationale on ATAC-seq analyses becomes clear.
3. SPT6 regulation on p63 gene expression has become more clear.
4. Although authors did not compare data with other stratified and non-stratified epithelial cells, as I suggested, they did perform functional analyses using intestinal organoids to show these SPT6 kd cells have intestinal cell-like properties. This is much appreciated. My original suggestion was not meant to ask authors to perform all the experiments but to use publicly available data.

Two minor suggestions:

- a) Authors emphasize that SPT6 regulates differentiation genes. However their data suggest that SPT6 does not only regulate differentiation genes but also genes important at the proliferation stage. P63 is a good example. In addition, when checking SPT6 ChIP-seq data in their Supplementary Table 5, many genes such as KRT5 and KRT14 with SPT binding sites are genes highly expressed in the basal cells and important in the proliferation stage. So I would suggest that authors consider to rephrase the role of SPT6 to regulate epidermal genes, or at least tune down and not to give readers the impression that SPT6 only regulates differentiation genes.
- b) Consider to mention SPT6 kd cells as 'resemble intestinal cells', or 'intestinal-like cells', rather than calling these cells to have an intestinal phenotype, especially in their abstract.

Reviewer #3 (Remarks to the Author):

This is a revised version of a manuscript I reviewed previously. The authors have done a nice job of responding to my and the other reviewers' criticisms and greatly improved the study. It is especially important to see their previous conclusions strengthened by new data on the PAF complex (knockdown of which resembles that of SPT6 in effects on differentiation-associated genes) and by bioinformatic analysis at least consistent with the idea that pause-regulated transcription factors might in turn be activating recruitment-limited (i.e., non-paused) downstream genes. I can now recommend publication in Nature Communications.

We again thank the reviewers for all their efforts with this manuscript. We are excited and pleased to see such positive comments and approval for publication. Below we have addressed the remaining minor suggestions from reviewer #2.

REVIEWERS' COMMENTS

Reviewer #1 (Remarks to the Author):

Authors have addressed all comments and concerns. I find the revised manuscript much improved and suitable for publication.

Reviewer #2 (Remarks to the Author):

This referee appreciates authors' great effort to address all questions raised by referees. Most of my concerns have been addressed and the manuscript has significantly improved. Several improvements relevant to my previous comments:

1. The manuscript has a better focus, shifted more on the mechanism of pol2 pausing and elongation. Especially including the new work on PAF1 has strengthen the manuscript.
2. The rationale on ATAC-seq analyses becomes clear.
3. SPT6 regulation on p63 gene expression has become more clear.
4. Although authors did not compare data with other stratified and non-stratified epithelial cells, as I suggested, they did perform functional analyses using intestinal organoids to show these SPT6 kd cells have intestinal cell-like properties. This is much appreciated. My original suggestion was not meant to ask authors to perform all the experiments but to use publicly available data.

Two minor suggestions:

- a) Authors emphasize that SPT6 regulates differentiation genes. However their data suggest that SPT6 does not only regulate differentiation genes but also genes important at the proliferation stage. P63 is a good example. In addition, when checking SPT6 ChIP-seq data in their Supplementary Table 5, many genes such as KRT5 and KRT14 with SPT binding sites are genes highly expressed in the basal cells and important in the proliferation stage. So I would suggest that authors consider to rephrase the role of SPT6 to regulate epidermal genes, or at least tune down and not to give readers the impression that SPT6 only regulates differentiation genes.
- b) Consider to mention SPT6 kd cells as 'resemble intestinal cells', or 'intestinal-like cells', rather than calling these cells to have an intestinal phenotype, especially in their abstract.

A) We have now added a paragraph (underlined text) in the discussion section to let the readers know that SPT6 can bind to basal layer genes however the vast majority of the genes bound by SPT6 are indeed epidermal differentiation ones. We have phrased this as follows:

It should be noted that SPT6 also binds and regulates the expression of basal layer specific genes such as KRT5, P63, and KRT14 (Supplementary Tables 4-5). However they are a small fraction in comparison to the epidermal differentiation genes that SPT6 binds and regulates.

B) We have changed intestinal phenotype to intestinal-like phenotype

Reviewer #3 (Remarks to the Author):

This is a revised version of a manuscript I reviewed previously. The authors have done a nice job of responding to my and the other reviewers' criticisms and greatly improved the study. It is especially important to see their previous conclusions strengthened by new data on the PAF complex (knockdown of which resembles that of SPT6 in effects on differentiation-associated genes) and by bioinformatic analysis at least consistent with the idea that pause-regulated transcription factors might in turn be activating recruitment-limited (i.e., non-paused) downstream genes. I can now recommend publication in Nature Communications.